# Multi-modal molecular programs regulate melanoma cell state

Miles C. Andrews [1,2,15], Junna Oba [3,4,15], Chang-Jiun Wu [5,15], Haifeng Zhu[3,15], Tatiana Karpinets[5], Caitlin A. Creasy[3], Marie-Andrée Forget[3], Xiaoxing Yu[4], Xingzhi Song[5], Xizeng Mao[5], A. Gordon Robertson[6,7], Gabriele Romano[8], Peng Li[8], Elizabeth M. Burton[5], Yiling Lu[5], Robert Szczepaniak Sloane[2], Khalida M. Wani[8], Kunal Rai [5], Alexander J. Lazar [5,8,9], Lauren E. Haydu [2], Matias A. Bustos [10], Jianjun Shen[11], Yueping Chen[11], Margaret B. Morgan[12], Jennifer A. Wargo [2,5], Lawrence N. Kwong[8], Cara L. Haymaker [8], Elizabeth A. Grimm[3], Patrick Hwu[3,13], Dave S. B. Hoon [10], Jianhua Zhang [5], Jeffrey E. Gershenwald [2], Michael A. Davies [3], P. Andrew Futreal [5], Chantale Bernatchez [3,14] & Scott E. Woodman [3,5✉]

Melanoma cells display distinct intrinsic phenotypic states. Here, we seek to characterize the molecular regulation of these states using multi-omic analyses of whole exome, transcriptome, microRNA, long non-coding RNA and DNA methylation data together with reverse-phase protein array data on a panel of 68 highly annotated early passage melanoma cell lines. We demonstrate that clearly defined cancer cell intrinsic transcriptomic programs are maintained in melanoma cells ex vivo and remain highly conserved within melanoma tumors, are associated with distinct immune features within tumors, and differentially correlate with checkpoint inhibitor and adoptive T cell therapy efficacy. Through integrative analyses we demonstrate highly complex multi-omic regulation of melanoma cell intrinsic programs that provide key insights into the molecular maintenance of phenotypic states. These findings have implications for cancer biology and the identification of new therapeutic strategies. Further, these deeply characterized cell lines will serve as an invaluable resource for future research in the field.

[1] Department of Medicine, Monash University, Melbourne, VIC, Australia. [2] Department of Surgical Oncology, The University of Texas MD Anderson Cancer Center, Houston, TX, USA. [3] Department of Melanoma Medical Oncology, The University of Texas MD Anderson Cancer Center, Houston, TX, USA. [4] Department of Extended Intelligence for Medicine, The Ishii-Ishibashi Laboratory, Keio University School of Medicine, Tokyo, Japan. [5] Department of Genomic Medicine, The University of Texas MD Anderson Cancer Center, Houston, TX, USA. [6] Canada's Michael Smith Genome Sciences Center, BC Cancer, Vancouver, BC, Canada. [7] Dxige Research Inc., Courtenay, BC, Canada. [8] Department of Translational Molecular Pathology, The University of Texas MD Anderson Cancer Center, Houston, TX, USA. [9] Department of Pathology, The University of Texas MD Anderson Cancer Center, Houston, TX, USA. [10] Departments of Translational Molecular Medicine and Genomic Sequencing Center, St John's Cancer Institute, Providence Saint John's Health Center, Santa Monica, CA, USA. [11] Department of Epigenetics and Molecular Carcinogenesis, The University of Texas MD Anderson Cancer Center, Smithville, TX, USA. [12] Sheikh Khalifa Bin Zayed Al Nahyan Institute for Personalized Cancer Therapy, The University of Texas MD Anderson Cancer Center, Houston, TX, USA. [13] H Lee Moffitt Cancer Center, Tampa, FL, USA. [14] Department of Biologics Development, Division of Therapeutics Discovery, The University of Texas MD Anderson Cancer Center, Houston, TX, USA. [15]These authors contributed equally: Miles C. Andrews, Junna Oba, Chang-Jiun Wu, Haifeng Zhu.
✉email: swoodman@mdanderson.org

Early gene expression profiling of metastatic melanoma tumors demonstrated that melanoma cells display distinct gene expression signatures, corresponding broadly with either a "proliferative" or an "invasive" in vitro phenotype[1,2]. Expression of melanocyte-inducing transcription factor (MITF) maintains a more differentiated melanocytic phenotype, associated with a set of features including melanin production, expression of melanocytic antigens, and proliferation[3,4]. Conversely, the receptor tyrosine kinase AXL has become a prototypic gene expressed at high levels in the invasive or "de-differentiated" state, characterized also by high expression of ZEB1 and low expression of MITF[5]. Single-cell methods confirm that both subtypes of melanoma cells can co-exist within tumors[6], and phenotypic switching between these two subtypes can occur[2] accompanied by transcriptional changes that resemble the epithelial-to-mesenchymal transition (EMT)[7].

The layers of molecular regulation that control cellular phenotypes and EMT-like processes in melanoma, and their relationships to therapeutic outcome, are only partly understood. Multiple in vitro studies have shown that MITF-high melanoma cells are more sensitive to small-molecule BRAF inhibition than MITF-low cells[8–11], but the connection between melanoma cell phenotype and BRAF inhibitor sensitivity is affected by the duration of drug exposure[12,13], co-occurring resistance aberrations, and in vivo environmental factors[14–16]. The implications of melanoma cell phenotypes for immunotherapy are also not yet clear. Therapeutic antibodies that block inhibitory immune checkpoint receptors or their ligands (CTLA-4, PD-1, PD-L1; collectively termed immune checkpoint inhibitor (CPI) therapies) have shown dramatic results in the treatment of melanoma and other cancers[17–20] achieving durable responses, but only in a minority of patients[21]. We, and others, have shown that a variety of factors influence responsiveness to CPI, including tumor mutation burden[22–25], somatic copy-number alterations (CNA)[25,26], antigen processing and presentation[27], CD8+ T cell tumor infiltration[28,29], presence of B cells and tertiary lymphoid structures[30,31], spatial relationships between immune and cancer cells[32], presence of specific functional immune cell subsets[33], and characteristics of the gut microbiota[34–36]. Notably, no single feature accounts for response or resistance in all cases.

In this work, in order to identify key molecular features that regulate melanoma cell transcriptomic states (hereafter "MCS") and corresponding phenotypes, we perform a large-scale integrative assessment of mRNA, microRNA mature strand (miR) and long non-coding RNA (lncRNA) expression, somatic mutations, CNA, and DNA methylation profiles across highly pure, patient-derived melanoma cell lines. We identify concurrent and multi-modal molecular interactions that underlie three distinct MCS, and find evidence for relative molecular stability of the most prevalent melanocytic/differentiated state. Furthermore, we show that the MCS-defining gene sets can also be used to infer MCS states of unsorted melanoma tumor specimens, demonstrate correlations with distinct tumor immune infiltrates, and impact treatment responses.

## Results

**Ex vivo gene expression subtypes incorporate a melanoma lineage spectrum.** We performed gene expression profiling of 68 early passage melanoma cell lines ("MDACC cell lines") to characterize transcriptomic signatures originating specifically from melanoma cells, without accompanying stromal and immune elements. In total, 68 cell lines derived from distinct metastatic tumors of 62 patients underwent RNA sequencing (RNA-Seq), broadly surveying melanomas of varying subtypes and clinical backgrounds (Supplementary Data 1). Three distinct transcriptomic subtypes were identified by unsupervised consensus clustering using non-negative matrix factorization (NMF) of the top 1500 most-variant genes (Supplementary Fig. 1a–c). Differential gene expression analysis identified 98 genes upregulated in cluster 1 (Supplementary Data 2, Fig. 1a), which we termed the melanocytic-like (MEL) set due to the presence of classic melanocytic markers (e.g. MITF, MLANA, TYR, DCT, SILV, OCA2, SOX10), and significant gene ontology (GO) enrichment for melanin biosynthetic process, melanocyte differentiation (false-discovery rate (FDR) < 0.005), and enrichment of the melanogenesis KEGG pathway (FDR < 0.10) (Supplementary Data 3). Similarly, 149 genes were upregulated in cluster 3, which we termed the mesenchymal-like (MES) set (Supplementary Data 2) due to the inclusion of classic EMT-implicated genes (e.g. ZEB1, AXL, ADAM12, COL1A1/5A1/6A2), GO enrichments encompassing extracellular matrix organization, cell adhesion, and cell migration (FDR < 0.00001), and enrichments of focal adhesion and PI3K-AKT signaling KEGG pathways (FDR < 0.0005), consistent with a more motile/invasive phenotype (Supplementary Data 3).

**Mixed neural and plasticity characteristics define the non-MEL/non-MES state.** Melanoma cell lines with high expression of MEL or MES genes are referred to being in the MEL or MES states, respectively. The presence of a third set of clustered samples (cluster 2) was superficially consistent with a transitional state in the phenotype-switching model of melanoma previously proposed by others[1,2,37,38]. Compared with MEL and MES state samples, cluster 2 samples had significantly enriched expression of 78 genes, and decreased expression of 6 genes (Supplementary Data 2), with KEGG pathway enrichments (pooling all differentially expressed genes) for sphingolipid metabolism, axon guidance, JAK-STAT signaling pathway, VEGF signaling pathway, BCR signaling pathway, focal adhesion, apoptosis, pancreatic cancer, small cell lung cancer (all $q = 0.061$), and TCR signaling pathway ($q = 0.085$)(Supplementary Data 3).

To characterize cluster 2 further, we used the SubMap algorithm[39] to compare cluster assignments between our cell lines and a previous transcriptomic analysis of melanoma; the MEL state was highly and specifically associated with the "melanocytic" state described by Tsoi et al.[38], while the MES state was highly associated with the "undifferentiated" state, and weakly with the "neural crest like" state. Cluster 2 was highly associated with both the "neural crest like" and "transitory" states of Tsoi (Supplementary Fig. 1d), and cluster 2 enriched genes showed overlaps of at least 5% exclusively with the "transitory", "transitory-neural crest cells" and "neural crest cells" signatures. Genes downregulated in cluster 2 samples showed a minor but significant overlap only with the "melanocytic" signature (Supplementary Fig. 1e). Overall, these findings support the assignment of a hybrid transitory state with neural crest features to our cluster 2, which we thus call a "neural-plastic" (NPLAS) state. This was additionally supported by gene set enrichment analysis of genes specifically enriched/depleted in our three-cluster model with the signatures proposed by Rambow et al.[40], based on single-cell sequencing analyses of melanoma xenografts. The MEL gene set was enriched for the Rambow MITF, mitosis, pigmentation, and immune signatures ($q < 0.001$), and the hypometabolic signature ($q < 0.005$), consistent with the interpretation of the MEL state samples as being more melanocytic/differentiated and proliferative. Differentially expressed genes in NPLAS samples were enriched for the Rambow neuro signature ($q < 0.001$), supporting our designation of these samples. Finally, the MES state samples were enriched for the Rambow invasion signature ($q < 0.001$), also consistent with the known invasive/motile potential of EMT-shifted cell states.

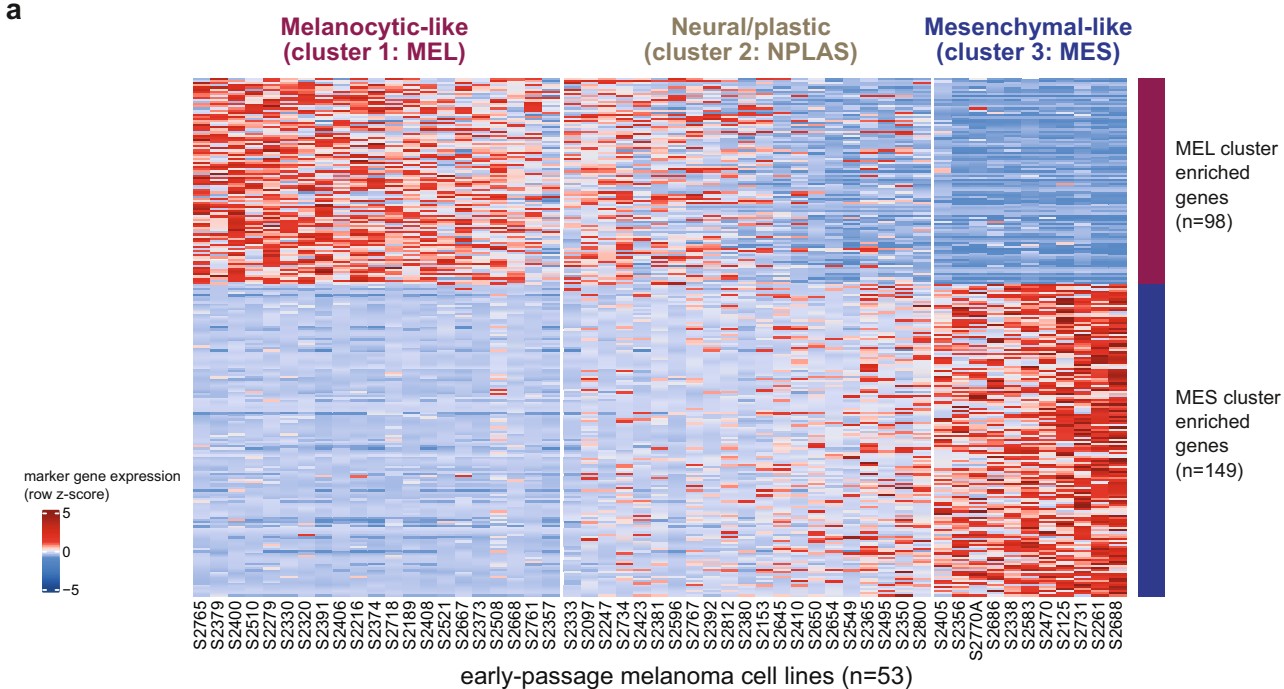

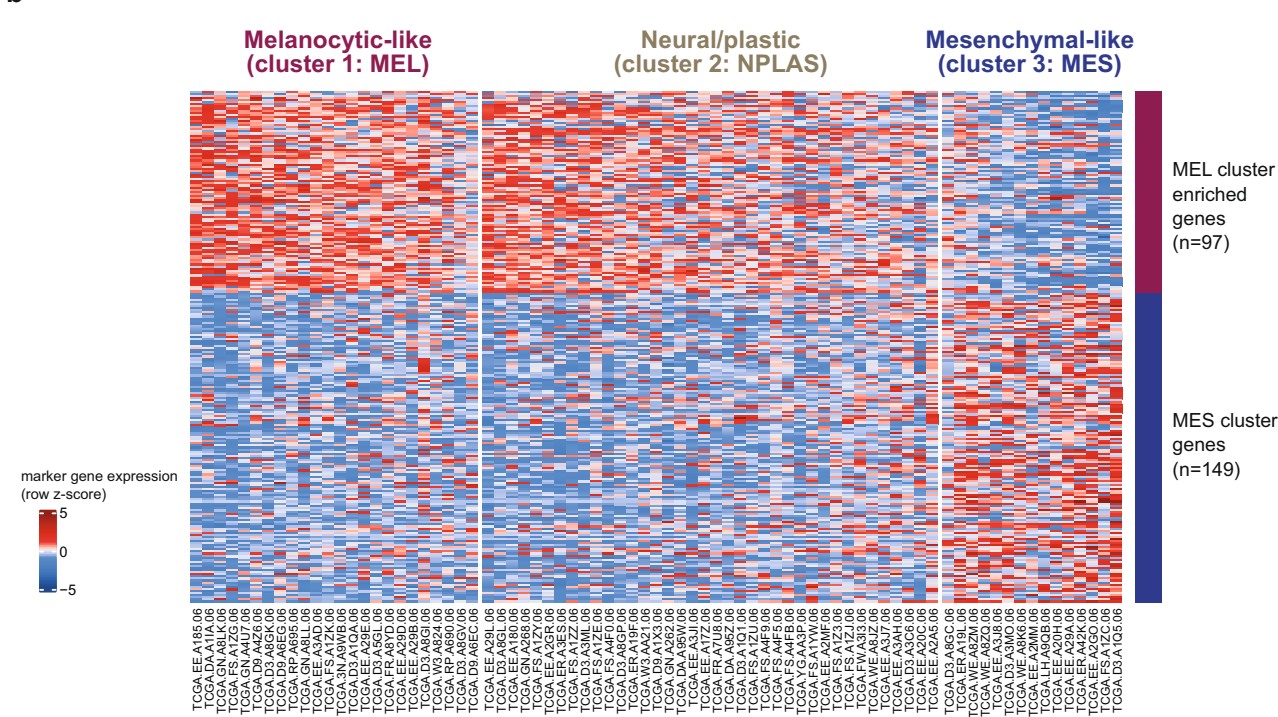

**Fig. 1 Differentially expressed genes between MEL and MES clusters.** Genes that were differentially expressed ($p < 0.05$) between MEL and MES samples. **a** Early passage melanoma MDACC cell lines ($n = 53$, see also Supplementary Fig. 2b, d). **b** High-purity metastatic TCGA melanoma samples ($n = 77$, purity = 0.85 by ABSOLUTE and/or CPE) with cluster assignments by cNMF clustering based on MEL and MES gene expression (one MEL gene was unavailable in TCGA data). In both panels, samples are ordered (left to right) within each cluster by decreasing the MEL-MES score (see "Methods"). Source data are provided as a Source Data file.

**Common biology underpins published gene sets**. We also evaluated the similarity of our MEL and MES gene sets with several published transcriptomic signatures derived from melanoma microarray analyses (Hoek et al.[2]), sequencing (Verfaillie et al.[41]) and single-cell studies (Tirosh et al.[6]) (Supplementary Fig. 1f). Overlaps with the Tirosh set were heavily driven by MEL genes; overlaps with the Hoek gene set were fairly balanced for MEL and MES representation, consistent with both gene set discovery strategies considering the extremes of the phenotypes for the purposes of differential expression. MEL and MES gene overlaps with the Verfaillie gene sets were also balanced, notwithstanding the lack of overlap for the vast majority of Verfaillie signature genes with any other signature due to their large size and content of non-coding RNA species. Re-clustering of our melanoma cell line panel based on the expression of all these gene sets excluding non-coding, deprecated, or pseudogene content demonstrated overall good consensus (Supplementary Fig. 2a), consistent with a common underlying biology driving sample clusters, rather than gene signature redundancy given the relatively minimal overlap of specific genes included in these signatures.

In general, melanoma cells derived from the same patient (11 samples from 5 patients) clustered closely together, and the few lower purity samples were consistently placed within the MES state (Supplementary Fig. 2b). To enable the application of the MCS as defined in our cell line panel to other datasets, we trained a random forest MCS classifier algorithm on our melanoma cell lines using cNMF cluster assignments as the classification input (see Methods; top 20 most informative genes by the variable importance metric shown in Supplementary Fig. 2c; retained genes in the final model shown in Supplementary Data 4). Subsequent analyses were performed on a refined subset of 53 cell lines after removing samples of lower purity or non-cutaneous melanoma subtypes, and retaining only one distinct cell line per individual source patient (Fig. 1a, details provided in Supplementary Fig. 2b, d).

**MCS can be inferred from bulk tumor gene expression data**. To determine if the gene sets that defined the MCS in highly pure melanoma populations ex vivo could also identify clusters in melanoma tumors, we examined their expression in metastatic samples from The Cancer Genome Atlas skin cutaneous melanoma (SKCM) dataset (hereafter "TCGA melanoma")[42]. Within high tumor purity metastases (cancer cell fraction ≥ 0.85 (proportion of nuclei), mean purity = 0.91 (CPE, see "Methods"); $n = 77$), 24 (31%) and 15 (19%) of samples displayed predominantly MEL or MES gene expression, respectively, and the remaining 38 (49%) clustered in a third, less-specific group analogous to the NPLAS group in cell lines (Fig. 1b). When considering all metastatic tumors regardless of tumor purity (% tumor nuclei ≥ 0.50, mean purity = 0.70 (CPE, see "Methods"); $n = 368$), similar patterns of MEL or MES gene expression were observed, albeit with less precise delineation between MEL and NPLAS gene expression patterns, as would be expected in a sample cohort of lower average tumor purity (Supplementary Fig. 3a). Notably, unlike the unsupervised cNMF approach, the random forest MCS classifier appeared highly conservative when classifying samples as MES, largely assigning predominantly MES gene-expressing samples as NPLAS, suggesting that in the context of bulk (i.e. multicellular) tumors, gene expression data contains signals that confound the ability of the classifier to identify MES samples. Nevertheless, despite these apparent limitations in classifying some individual samples, the MEL and MES gene sets can be used broadly to infer the MCS of melanoma cells contained within the whole (bulk) tumors.

**MCS is associated with tumor immune content**. To determine whether MCS influences tumor immune content, we categorized all metastatic TCGA melanoma samples using cNMF clustering based on the expression of our MEL and MES gene sets. We then estimated the content of major immune and stromal cell types from gene expression data with the MCP-counter algorithm[43] and observed substantial variation in immune composition between the three MCS groups (Fig. 2a). When immune infiltrated, MEL samples displayed generally lymphoid/monocytic content, MES samples displayed high stromal (endothelial, fibroblast) and/or myeloid/monocytic content, and NPLAS samples tended to have the least multi-lineage immune representation (Fig. 2a–c). Some MEL samples nevertheless displayed marked enrichment of cytolytic markers such as *PRF1* and *GZMA* (Fig. 2d). The median expression levels of most lymphocytic marker genes varied across MCS groups, including regulatory molecules *FOXP3*, *PDCD1*, *CTLA4,* and *LAG3*, with the lowest expression observed in NPLAS samples, (all $q < 1e{-}03$; Supplementary Fig. 3b). MES samples displayed significantly higher expression of the M2-macrophage markers *MRC1* (CD206), *CD163*, and *IL10* compared to other MCS (all $q < 1e{-}10$; Fig. 2e), a cell population which has been implicated in driving the formation of a cancer-promoting tumor microenvironment across multiple cancer types[44,45].

**Stromal signatures confound interpretation of MCS in MES samples**. Of the gene sets used by MCP-counter to enumerate 10 non-cancer cell populations, only the fibroblast (3 of 8 genes shared; *COL1A1*, *COL6A1*, *COL6A2*) and endothelial (2 of 33 genes shared; *ESM1*, *HECW2*) gene sets exhibited overlap with the MES gene set, and no overlaps with the MEL gene set were observed. Furthermore, we applied four widely used immune deconvolution algorithms (EPIC[46], quanTIseq[47], MCP-counter[43], and TIMER[48]) to our cell line data for which non-tumor content is minimal or absent, and confirmed that estimation of stromal and myeloid signatures are highly inaccurate in melanoma samples, being particularly unreliable for estimation of cancer-associated fibroblast scores in MES cluster melanoma samples (Supplementary Fig. 4). Tumor purity was not identical across MCS subtypes of the TCGA melanoma samples, although was more similar by histologic assessment (median percent tumor cells by histology; 85% MEL, 90% NPLAS, 85% MES; $p = 4.8e{-}05$, Kruskal–Wallis test) than by genomic measures for which MES samples scored lowest (median purity estimate by CPE method; 0.79 MEL, 0.82 NPLAS, 0.66 MES; $p = 2.2e{-}16$, Kruskal–Wallis test). Together, these data indicate that overlapping gene expression profiles between MES state melanoma cells and stromal elements, and potentially in some cases lower tumor purity, confound the interpretation of stromal deconvolution in melanoma tumors, if based on gene expression alone.

**MCS gene expression patterns indicate complex underlying regulation**. Expression of MEL/MES genes in our melanoma cell lines revealed distinct patterns suggesting the culmination of multiple regulatory inputs, including graded (linear) relationships for canonical marker genes such as *MITF* (MEL) and *ZEB1* and *AXL* (MES) or switch-like expression (abrupt and non-overlapping "on" or "off" levels of expression) for genes such as *SOX10* and *EDNRB* (MEL) and *EGFR* (MES) (Fig. 3a). To understand how these MCS-defining genes achieve complex expression patterns, we next analyzed the same melanoma cell lines for somatic mutations, CNA, DNA methylation, protein levels, and miR and lncRNA expression (Supplementary Data 5 and 6).

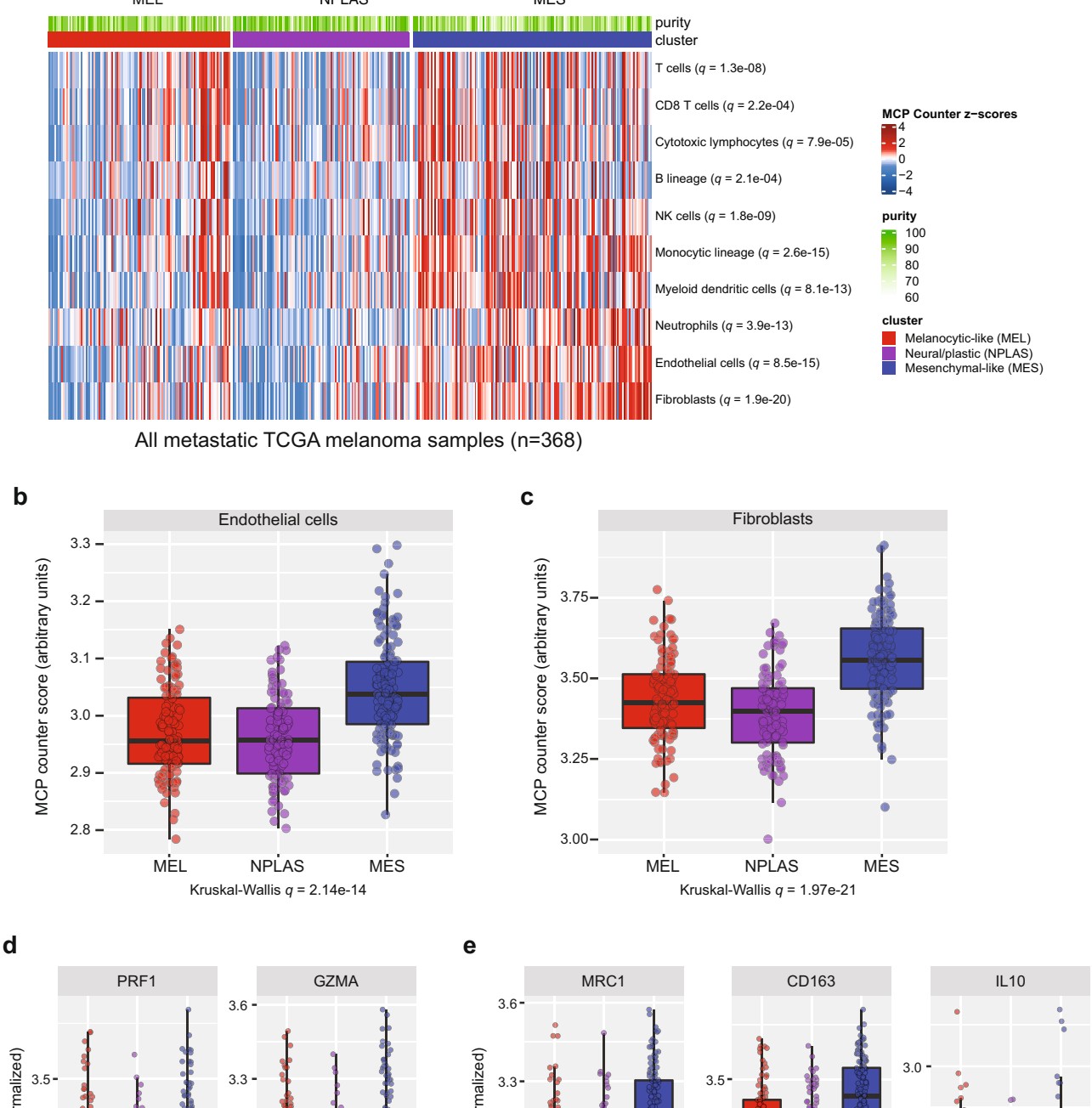

**MCS are not driven by mutational differences**. Among a diverse landscape of mutations and CNA affecting known oncogenes and tumor suppressor genes in melanoma[49] (Fig. 3b), the most frequently altered genes were *BRAF* (73%), *CDKN2A* (71%), *CDKN2B* (57%), *PTEN* (55%), and *TP53* (55%), with high prevalence of *NRAS* mutations in this cohort (41%). There was little association between recurrently altered genes and MCS;

however, we did observe a higher frequency of *BRAF* hotspot events in the MEL samples ($p = 0.016$, Fisher's exact test) and of *NRAS* hotspot events in the MES samples ($p = 0.03$, Fisher's exact test), consistent with the typically mutually exclusive alteration of these two genes. In high-purity TCGA melanoma samples, *BRAF* mutations were not associated with MCS (Supplementary Fig. 5); thus, we cannot exclude a potential selective

**Fig. 2 MCS are associated with differences in tumor immune content. a** Heatmap of immune and stromal cell type composition in metastatic TCGA melanoma samples of any tumor purity ($n = 368$) derived using MCP-counter transcriptomic deconvolution. Also indicated are tumor cell purity estimates derived from histologic examination as provided in the TCGA melanoma metadata (% tumor nuclei, range 60–100%), and the MCS membership defined by cNMF clustering of samples using MEL and MES gene expression. Statistical significances are given as Benjamini-Hochberg corrected $q$ values, from two-sided Kruskal–Wallis tests for differences across the three clusters. In the same metastatic TCGA melanoma samples ($n = 368$), distributions of (**b**) MCP-counter endothelial cell signature scores, and (**c**) MCP-counter fibroblast signature scores, (**d**) cytolytic markers *PRF1* and *GZMA*, (**e**) key M2-macrophage marker genes *MRC1*, *CD163* and *IL10*. In (**b**–**e**), Benjamini–Hochberg corrected $q$ values from two-sided Kruskal–Wallis tests across all three sample classes are shown. Boxplots indicate the median (thick bar), first and third quartiles (lower and upper bounds of the box, respectively), lowest and highest data value within 1.5 times the interquartile range (lower and upper bounds of the whisker), and all individual data points are shown. Source data are provided as a Source Data file.

advantage for *BRAF* mutated MEL samples when grown in vitro. At the nucleotide level, mutations were dominated by C>T transitions, as is typical for the UV signature in melanoma; no clear mutational signature differences were evident between the three MCS (Fig. 3b).

**MCS display distinct patterns of CNA.** In contrast to the mutational landscape, distinct CNA differences were observed between MCS, with MEL samples showing relative gains of chromosomes 1p, 7q and 15 (p + q), and losses of chromosomes 9p, 11 (p + q), 18p and 19q (Fig. 3c). As a group, MEL genes were more commonly affected by gene-level CNA than MES genes (32 of 98 (33%) MEL vs 35 of 149 (23%) MES genes, Chi-square test $p = 0.15$; Supplementary Data 6). Furthermore, CNAs affecting MES genes were typically driven by gene-level losses in MEL or NPLAS samples rather than gains in MES samples, consistent with CNA events (of either direction) primarily occurring in MEL cells.

Dissimilar global copy-number profiles across the three MCS, particularly affecting chromosomes 7q, 8q, 10, and 18 (red asterisks in Fig. 3c), appeared inconsistent with the NPLAS state simply being comprised of a combination of the MEL and MES profiles, as would be expected for a truly transitional or mixed state lying midway on a MEL-to-MES spectrum. We then examined this relationship in melanoma cell lines of the Cancer Cell Line Encyclopedia (CCLE), using RNA-Seq data and cNMF clustering to define each sample's MCS (Supplementary Fig. 6a) followed by a comparison of genome-wide segmented CNA profiles across each cluster (Supplementary Fig. 6b). As in our own cell lines, CNA profiles of NPLAS samples displayed regions of apparently distinct CNA (e.g. 2p, 5p, 9q, 12q, 14p, 18) inconsistent with these samples representing mixtures or an alternative phenotypic state of cells harboring the MEL and MES profiles, supporting the existence of a distinct NPLAS state. Importantly, regional CNA observed in CCLE samples were not directly equivalent to those seen in our early passage cell lines; previous studies[50] have demonstrated the evolution of copy-number profiles in samples following extended passaging in vitro, which may explain some of these differences.

**DNA methylation associates with MCS.** We next identified numerous strong DNA methylation associations with MEL and MES gene expression, acting variably at canonical promoters/ enhancers, or gene-body sites. Transcript levels of conventional melanocytic markers (e.g. *SILV*, *MLANA*, and *TYR*) which are known to be dominantly regulated by MITF[4,51], and of *MITF* itself were significantly inversely correlated with DNA methylation in the cell lines and TCGA melanoma samples (Fig. 4a, Supplementary Data 6–8). Expression of melanocytic differentiation markers (e.g. *MLANA*, *TYR*, *TRPM1*, *DCT*) was widely variable in NPLAS samples, and DNA methylation was frequently associated with low expression, similar to that found in MES samples (Supplementary

Data 7–8). Although DNA hypomethylation of EMT-upregulated genes *AXL* and *KRT8/18*[52] was observed in MES samples, other MES genes were positively correlated with DNA methylation, including *EGFR*, *EPHB2*, *WNT5A*, and *SERPINB7* (Fig. 4b, Supplementary Data 6, 9, 10). In some cases, DNA methylation at different regions of the same gene had discordant associations with transcript abundance. For example, promoter methylation of *ZEB1* was associated with low *ZEB1* expression (MEL samples) and gene-body methylation was significantly correlated with high *ZEB1* expression (MES samples) (Fig. 4b, Supplementary Data 6, 9, 10). In vitro, MEL state cell lines displayed heightened sensitivity to the global hypomethylating agent decitabine (5-aza-2′-deoxycytidine) compared with MES state cell lines, suggesting a more specific cytotoxic/cytostatic effect in MEL state cells (Supplementary Fig. 7a). Whether this is due to greater dependence on specific methylation events in MEL cells or a skewed balance of hypomethylation to hypermethylation events between the MCS remains to be determined.

To determine if differences in epigenetic aging characterize the MCS, we evaluated the DNA methylation age (DNAmAge) estimates of our melanoma cell lines using the method of Horvath[53]. We found a very poor correlation between the chronological age of the patient at the time the source tumor was sampled and the calculated DNAmAge, similarly to what was originally described across multiple tumor types (Supplementary Fig. 7b). Despite high data dispersion, linear regression revealed highly similar trends for both MEL and MES groups for which DNAmAge was nearly constant regardless of chronological age (slope = −0.024 MEL, −0.023 MES). By contrast, the epigenetic age of NPLAS samples showed a positive correlation with chronological age (slope = 0.50). Together with distinct CNA profiles, these findings add further support to the existence of fundamental molecular differences between the NPLAS and either MEL or MES states.

**Differentially expressed miRs characterize MCS.** MiRs exert post-transcriptional regulation of gene expression. Thus, we performed whole-miRNome profiling using small RNA sequencing and found significant differential upregulation of 13 miRs in MEL and 9 miRs in MES cell lines (Fig. 5a and Supplementary Data 6). On a gene-by-gene basis, there was a consistent inverse correlation between MEL-associated miRs and MES genes, and vice versa in cell lines (Supplementary Fig. 7c); these associations were largely maintained in high-purity metastatic TCGA melanoma samples (Supplementary Fig. 7d and Supplementary Data 6). Minimal overlap was seen between genes affected by CNAs and genes showing evidence of regulation by miRs (Supplementary Data 6), suggesting additive regulatory potential.

Twelve of the 22 differentially expressed miRs had at least one MEL or MES gene in its set of known regulatory targets according to validated and predicted miR target databases (see "Methods"; Fig. 5b, c). The most highly differentially expressed miR of this group was miR-211-5p, a known melanoma tumor suppressor[54]

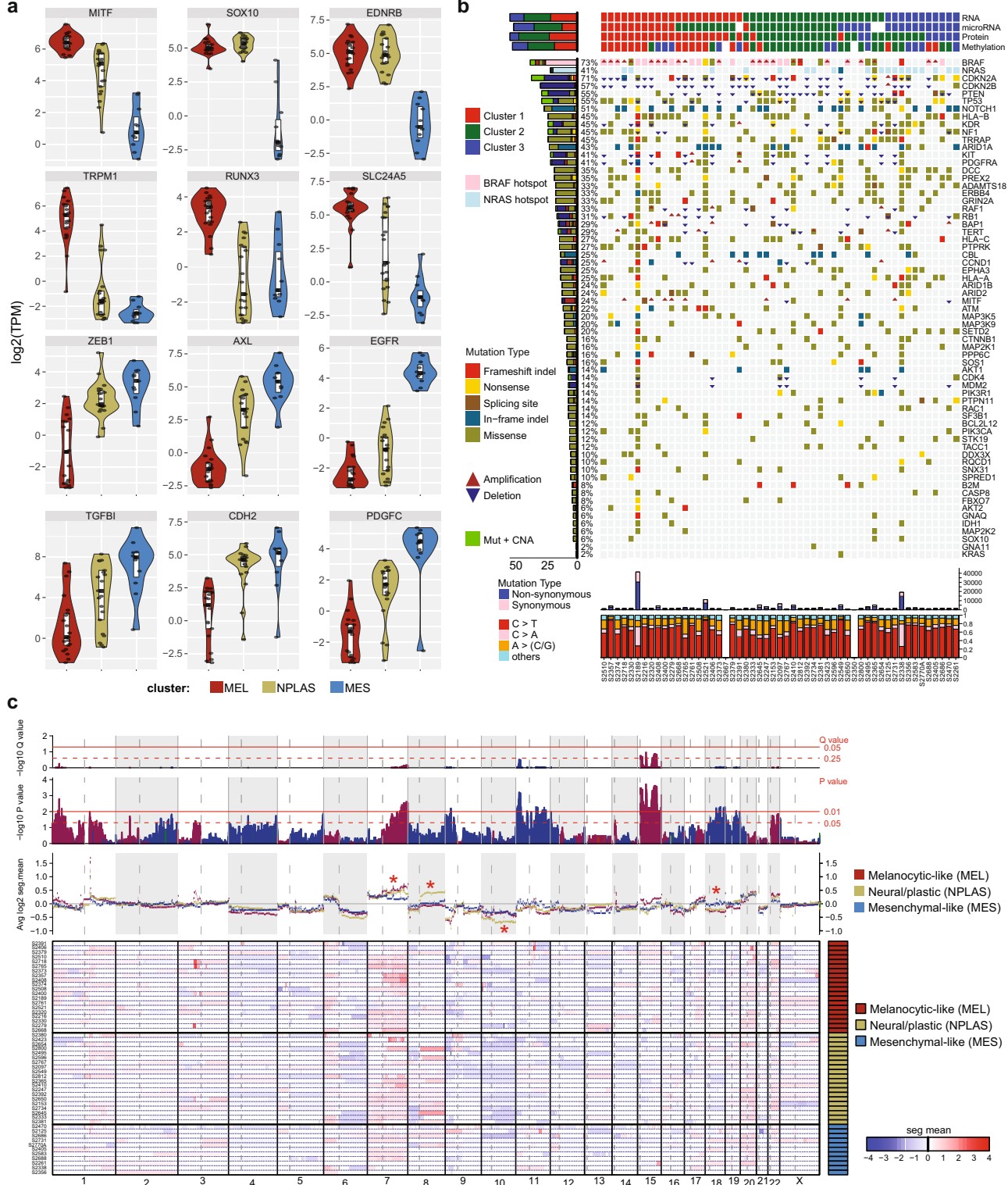

located within an intron of the *TRPM1* gene, a member of the MEL gene set. The coordinated upregulation of *TRPM1* and miR-211-5p in MEL samples was consistent with a shared transcriptional regulation, which in the case of miR-211-5p, reinforces MEL gene set expression due to miR targeting of several MES genes including *EPHB2*, *THSD4* (thrombospondin type I domain containing 4), *NUAK1* (cancer-promoting driver of EMT)[55], *KCNMA1* (cancer invasion and metastasis)[56], and *ZEB1* (a key driver of EMT-like processes in melanoma)[7]. Indeed, miR-211

can play tumor suppressive or promoting roles under different cell states and microenvironmental conditions in melanoma, at least partly dependent on target mRNA expression. We confirmed miR-211-5p targeting of *ZEB1* at both transcript and protein level and demonstrated knockdown of ZEB1 protein following transfection-mediated overexpression of miR-211-5p in melanoma cells with high endogenous levels of ZEB1 in vitro (Fig. 5d, e), supporting the validity of our approach to the identification of meaningful multi-modal regulatory relationships.

**Fig. 3 Relationships of mutations and DNA methylation to MEL or MES gene expression. a** Violin plots comparing expression of selected MEL and MES genes between cNMF-derived clusters in the early passage melanoma MDACC cell lines ($n = 53$) reveal several distinct types of relationship between gene expression and sample cluster type. These include: graded relationships (gradual transition in expression level across the MCS, e.g. *MITF*, *AXL*, *SLC24A5*); switch (non-overlapping high/low) relationships (e.g. *SOX10*, *EDNRB*); or genes almost exclusively expressed in one MCS only (e.g *EGFR*). Boxes indicate the median (thick bar), first and third quartiles (lower and upper bounds of the box, respectively). Violins (kernel probability density) extend to the lowest and highest data value. All individual data points are shown. **b** Oncomap of genomic alterations affecting a panel of genes commonly mutated or copy number altered in melanoma, with cNMF-derived MCS clusters indicated in the tracks above based on RNA expression, miR expression, protein expression (measured by RPPA), or DNA methylation pattern. Summary data of mutation type (synonymous or non-synonymous), and nucleotide changes are indicated below for each sample. The histogram at the left indicates the cumulative frequency of genomic events affecting each gene. **c** Genome-wide CNA indicated by heatmap and segmentation map, comparing results from MDACC cell lines grouped by cNMF-defined MCS. Source data are provided as a Source Data file.

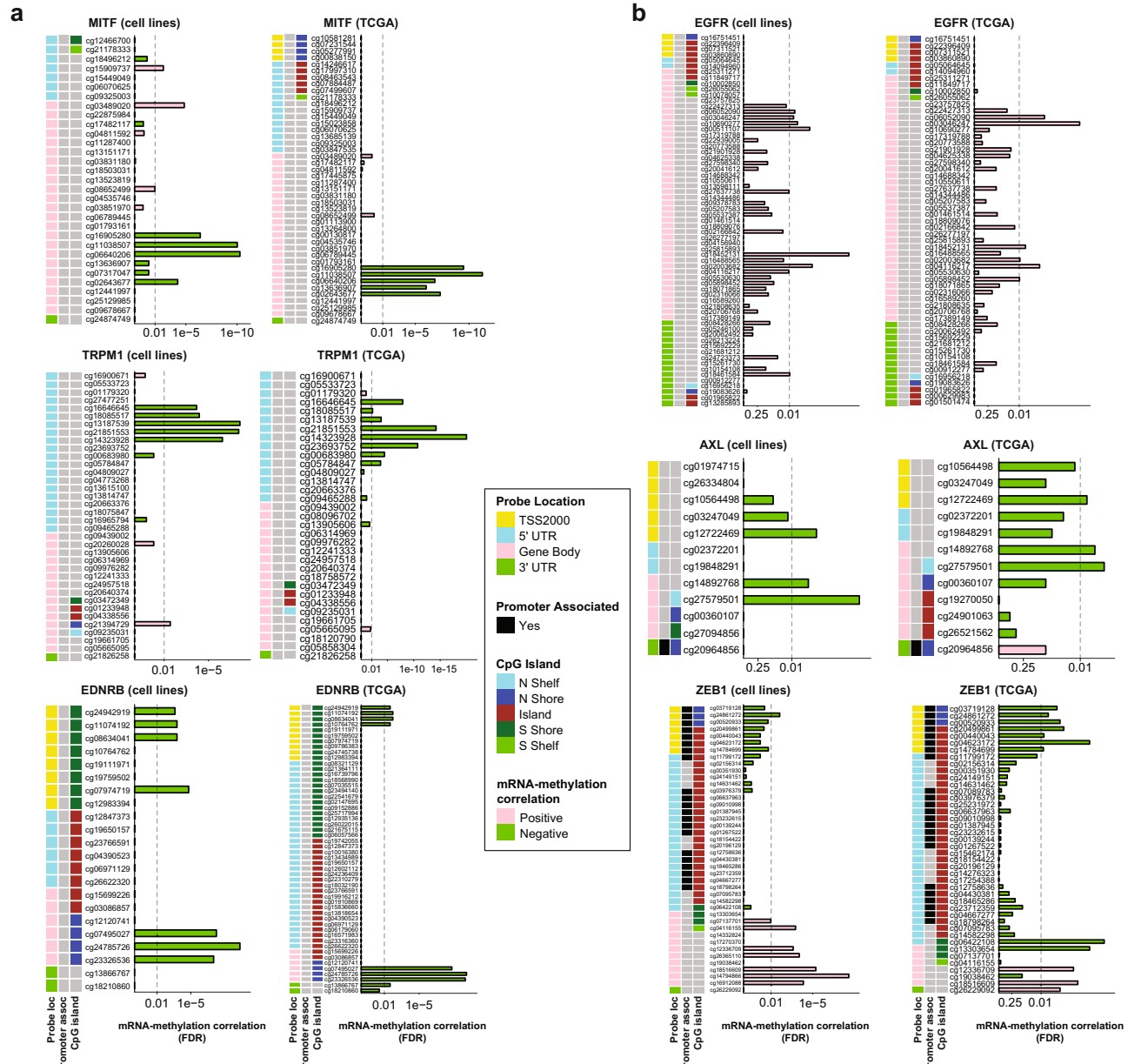

**Fig. 4 DNA methylation demonstrates site- and gene-specific associations with MEL and MES gene expression. a**, **b** Overview of statistically significant DNA methylation events for (**a**) 3 MEL genes and (**b**) 3 MES genes contrasting the observed exclusively negative (green bars), exclusively positive (pink bars), and mixed patterns of correlation between individual methylation probes and expression of the indicated gene. FDR-corrected two-sided Spearman correlation *p* values are shown. For each gene, DNA methylation in MDACC cell lines is shown at the left, and in TCGA melanoma samples at the right; note that due to processing pipelines used, the available datasets report overlapping but non-identical probesets per gene.

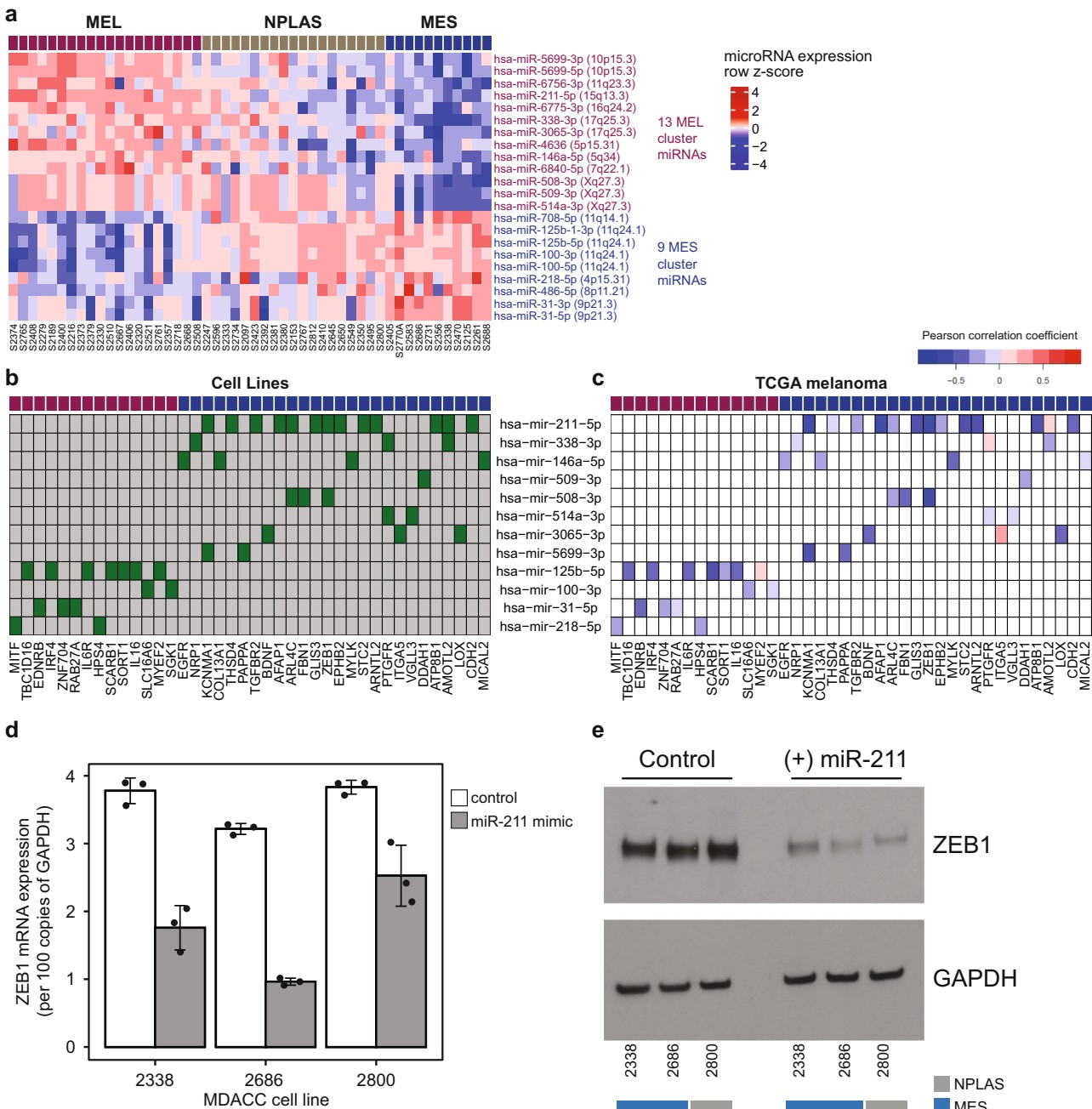

**Fig. 5 Inverse associations between MEL and MES gene expression and mature strand miRs in cell lines are preserved in melanoma tumors.**
**a** Heatmap depicting expression of miRs differentially expressed between MEL and MES early passage MDACC melanoma cell lines (*n* = 53). **b** Inverse Pearson correlations between abundances of miRs differentially expressed between MEL and MES samples, and the group of mRNAs confirmed by at least one experimentally validated and/or at least two in silico miR target databases to be targets of these miRs, in the early passage cell lines. More miRs targeting MES genes than MEL genes were identified; 8 active MES-targeting miRs (highly expressed in MEL samples) and 4 active MEL-targeting miRs (highly expressed in MES samples). **c** The majority of miR-mRNA associations identified in cell lines were also observed in TCGA melanoma tissue samples. **d** Effects of miR-211-5p overexpression in the cell lines indicated on *ZEB1* mRNA expression. Bars depict mean + SD of three biological replicates under control (white bars) or miR-211-5p mimic (gray bars) treatment conditions. **e** Representative Western blot of ZEB1 protein expression in the cell lines indicated comparing control and miR-211-5p overexpression conditions. Source data are provided as a Source Data file.

Levels of miR-31-5p were inversely associated with expression of its target MEL gene *EDNRB*, known to play an essential role in normal melanocyte development as well as melanoma proliferation, metastatic initiation, and BRAF inhibitor resistance[57–59]. Additionally, the mRNA levels of MEL genes *IRF4*, *IL16*, and *IL6R*, all of which have known immunomodulatory roles affecting lymphocyte activation, chemotaxis and differentiation, were diminished in cells expressing the MES-associated miR-125b-

5p, implicating miR-125b-5p as a pan-immune modifying miR in melanoma (Fig. 5b, c).

**Long non-coding RNAs are differentially expressed between MCS.** Several lncRNAs were highly differentially expressed between the MCS (Supplementary Fig. 8a–d). Compared to MES, MEL samples were enriched for *LINC00518*, a marker that

differentiates melanoma from nevi in gene signature analysis[60] and is associated with prognosis of uveal melanomas[61] (Supplementary Fig. 8a). Conversely, MEL samples had low levels of the hypoxia-inducible *H19* which may trigger a disseminating phenotype via *SNAIL* upregulation[62] (Supplementary Fig. 8c). Although absolute expression levels of these non-coding RNA species may be low, substantial signal amplification of effects on target gene expression may be a feature of such multi-step interactions. Detailed experimental evaluation of the lncRNA associations presented here will be required to elucidate the molecular mechanisms promoting specific MCS.

**Combined assessment of regulatory modalities suggests less intrinsic plasticity in the MEL state.** To compare multi-modal regulation between MCS further using the MDACC cell lines, we generated combined representations and summarized the three independent regulatory modes of CNA, DNA methylation and miR upon each member of the MEL and MES gene sets as directed categorical factors (−1 = significant inverse correlation, +1 = significant direct correlation, 0 = no significant correlation; see Methods) and clustered the genes based on similar patterns of multi-modal regulation (Fig. 6a, b; Supplementary Data 6). A small set of genes displayed all 3 modulatory features: MEL genes *MITF, TBC1D16, ENDRB, RAB27A, HPS4, SLC16A6, IL16,* and *SGK1*; and MES genes *AFAP1, ZEB1, GLIS3, MICAL2, ARNTL2, AMOTL2,* and *MYLK* (Fig. 6a, b; Supplementary Data 6). CNA typically co-occurred with at least one additional regulatory mode, although infrequently with miR regulation (67 CNA regulated; 48 (72%) DNA methylation co-regulated, 20 (30%) miRNA co-regulated). DNA methylation events were the most common modality of regulation occurring in isolation (109 of 247 genes, 44%; Fig. 6a, b). Overall, MEL genes were more frequently copy number altered than MES genes (32 of 98 (33%) MEL vs 35 of 149 (23%) MES genes) and a higher proportion of MES genes displayed evidence of regulation by non-CNA mechanisms (86 of 149 (58%) MES vs 43 of 98 (44%) MEL genes displaying non-CNA regulation; $p = 0.045$, Chi-squared test). MEL genes were significantly more affected by suppressive DNA methylation patterns (58 negative versus 9 positive mRNA-methylation correlations) than MES genes for which reinforcing DNA methylation patterns predominated (49 negative versus 58 positive correlations; MEL versus MES $p = 1.8\mathrm{e}{-07}$, Chi-squared test). Furthermore, genes whose expression is known to enforce the differentiated state (*SOX10* and *MITF*) were copy-number amplified, while genes that transition cells to a de-differentiated state (*AXL* and *ZEB1*) were allelic copy-number low in MEL samples (Supplementary Data 7, 9). Individual NPLAS samples typically expressed either *AXL* or *ZEB1*, whereas MES samples expressed both *AXL* and *ZEB1* at a high level, with high *EGFR* expression being a defining feature of this MCS and each of these genes being subject to multiple regulatory modalities (Fig. 6c). Conversely, the heavily regulated MEL gene *EDNRB* revealed frequent discordance between the expression of *MITF* and its transcriptional target gene *TRPM1*[63] as a defining feature of cells with an NPLAS phenotype (Fig. 6d). Notably, *TRPM1* and the miR-211 locus contained within its intron, are located on one of the most significantly differentially CNA regions in MEL samples (chromosome 15q13) (Fig. 3c) and are additionally subject to inhibitory DNA methylation patterns (Supplementary Data 7). Thus, a highly differentiated (MEL) state is associated with a higher burden of fixed structural aberrations, raising the possibility that this state is inherently less susceptible to perturbation by chemical or other microenvironmental signals, which may explain the higher prevalence of MEL samples in published cohorts of both cell lines and tumor samples[1,64]. Not all

regulatory associations identified in tumor cells (cell lines) remained identifiable in TCGA melanoma tumor samples (Supplementary Data 6), potentially due to genomic signal dilution by the diversity of cell types present even in high-purity tumor samples, but differences in the tissue sites of origin between the cell line and TCGA melanoma cohorts may have impacted these findings. However, the overall patterns observed at the cluster level (e.g. preponderance of CNA affecting MEL genes) were preserved.

**Clinical outcomes for melanoma sample donors are associated with MCS.** Many studies have explored the association between melanoma gene expression signatures and therapeutic outcomes[12,23,65,66] but are often limited by the timing of tumor sampling versus initiation of the treatment under study, or differences between experimental models. We leveraged the comprehensive clinical data of patients from whom our melanoma cell line cohort was derived to evaluate whether MCS were associated with clinical outcomes to the immediate next line of therapy received by the donor patients. Our cohort provides a wide survey of systemic treatment regimens including cytotoxic, targeted, and immune therapies (most common categories; TIL therapy, MAPK targeted therapy, biochemotherapy, each $n = 7$; Fig. 7a, Supplementary Data 1). Each of these groups showed trends for the lower duration of clinical benefit (defined as time from first dose until starting another line of therapy) in MES samples, although sample numbers were limiting for formal stratified analyses (Supplementary Fig. 9a). When considering all patients receiving subsequent therapy, regardless of treatment type, MES samples had a numerically lower duration of clinical benefit (MEL 172d [IQR 86-413d], NPLAS median 121d [IQR 79-180d], MES median 79d [IQR 34-97d]; Fig. 7b). Interestingly, this trend was also seen within NPLAS samples after subdivision into more MEL-like or more MES-like subgroups based on clustering strictly into two classes based on MEL/MES gene expression (Fig. 7b). As expected, MEL state cell lines harboring *BRAF*[V600E] mutations were markedly more sensitive to treatment with the BRAF inhibitor vemurafenib in vitro, experiencing near-complete loss of viability at 1 μM, in contrast to the highly resistant MES and NPLAS samples (Supplementary Fig. 9b).

**MEL state associates with favorable tumor immune microenvironments and depletion following CPI.** PD-1 CPI now forms the backbone of melanoma treatments but were not widely available at the time most of our cell lines were developed. Thus, to evaluate the relationship between MCS and response to PD-1 blockade we utilized a publicly available dataset of pre- and on-PD-1 CPI melanoma samples profiled for gene expression[67]. As expected, samples clearly separated into our three MCS based on MEL and MES gene expression (Fig. 7c), regardless of pre- or on-treatment time point and with inferred immune content for each cluster similar to that observed in TCGA melanoma samples which, notably, were treatment-naïve (Supplementary Fig. 10a, b). We assigned MCS classes to individual samples using a random forest classifier model trained on the MDACC cell line samples (see methods) and found a trend for significantly different RECIST-based objective responses (binary responder/non-responder) between MCS subgroups across the entire cohort ($p = 0.081$, Fisher exact test); however, this was driven by a significantly higher proportion of responders in the non-MEL subgroups at the on-treatment time point ($p = 0.0055$, Fisher exact test) as there was no significant difference in response by MCS in pre-treatment samples only ($p = 0.61$). When considering only cutaneous-type melanomas and samples with available response

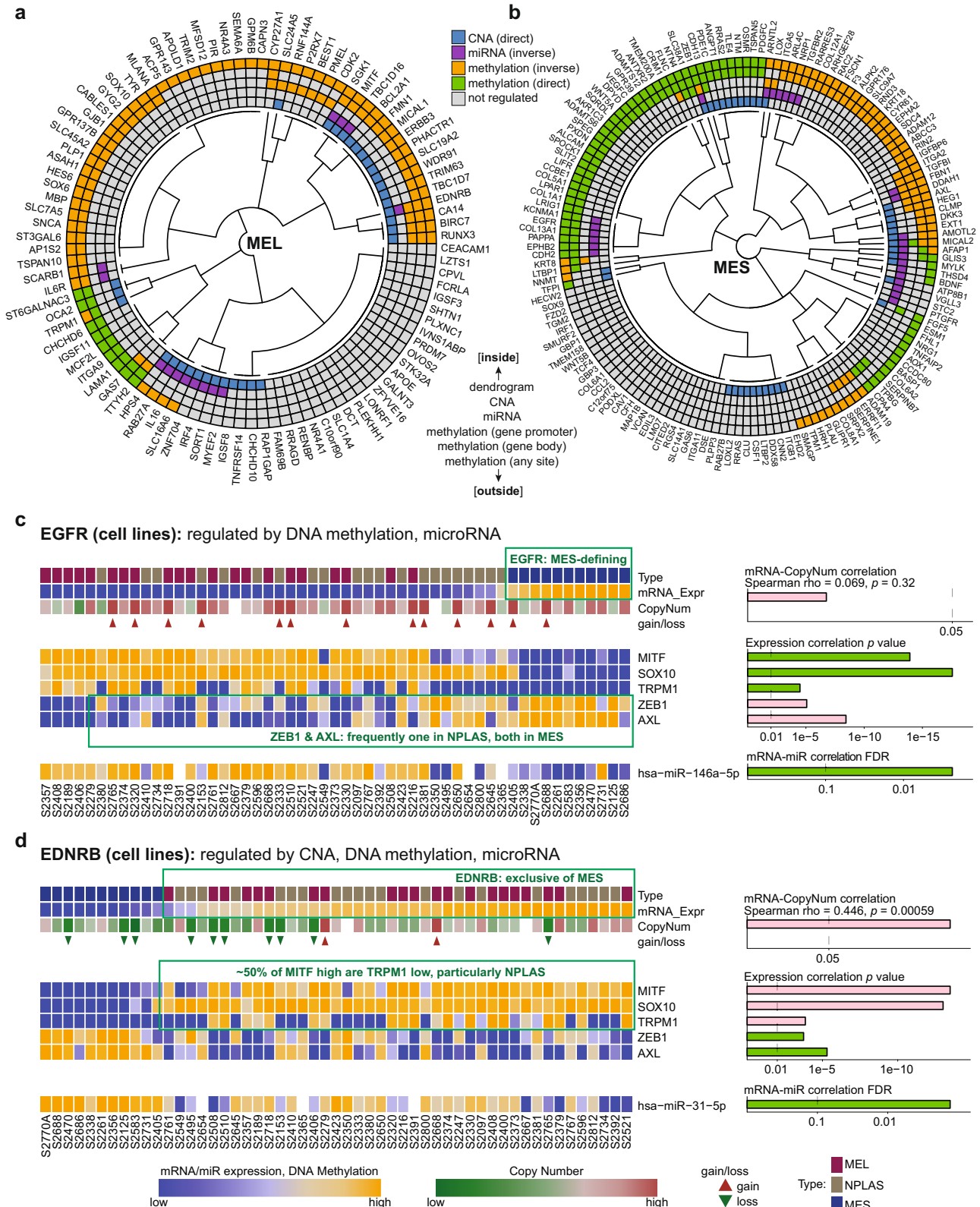

annotation, a strong trend for differential response by MCS remained at the on-treatment time point ($p = 0.051$).

Of the 42 patients with longitudinal samples and known response status, 9 displayed discordant MCS categories when we compared on-treatment with pre-treatment samples. A MEL-wise shift occurred in 2 patients (1 MES to NPLAS, 1 NPLAS to MEL) who were both non-responders. In contrast, a MES-wise shift occurred in 7 patients (2 MEL to NPLAS, 5 NPLAS to MES), comprised of 3 non-responders and 4 responders ($p = 0.069$, Fisher's exact test; Fig. 7d). These data require further corroboration using a larger cohort, but suggest that regardless of the starting MCS, patients who display a shift in MCS towards a more MES phenotype during PD-1 CPI treatment, may have a higher likelihood of response to this therapy.

**Fig. 6 Multi-modal regulatory influences on MEL and MES gene expression.** Regulatory associations with expression of the (**a**) MEL genes and (**b**) MES genes summarized (from inside to outside): CNA, miR expression, and DNA methylation at gene promoter, gene body, and any site. Genes are grouped according to similarity of active regulatory mechanisms, determined by unsupervised clustering using a Gower distance, as described in Methods. The direction of association between DNA methylation and gene expression is indicated by color (orange = inverse, green = direct). **c, d** Key molecular regulatory influences active on the hallmark genes (**c**) *EGFR* and (**d**) *EDNRB*, indicating key features affecting *ZEB1/AXL* and *MITF/TRPM1* in MES and MEL genes, respectively. From top down, annotation rows indicate: (in the upper block) MCS group ("Type"), gene expression ("mRNA_Expr"), copy number ("CopyNum"), copy-number gains/losses ("gain/loss"); (in the middle block) gene expression of *MITF*, *SOX10*, *TRPM1*, *ZEB1* and *AXL*; and (in the lower block) expression of the indicated significantly anti-correlated microRNA mature strands (miRs). Correlation statistics are indicated graphically to the right of each feature; one-sided (positively) Spearman correlation *p* values (mRNA-CopyNum) and FDR-corrected two-sided Pearson correlation *p* values (mRNA-miR). Samples are ordered left to right by expression of EGFR (**c**) or EDNRB (**d**). Missing data for individual samples are indicated by white. Source data are provided as a Source Data file.

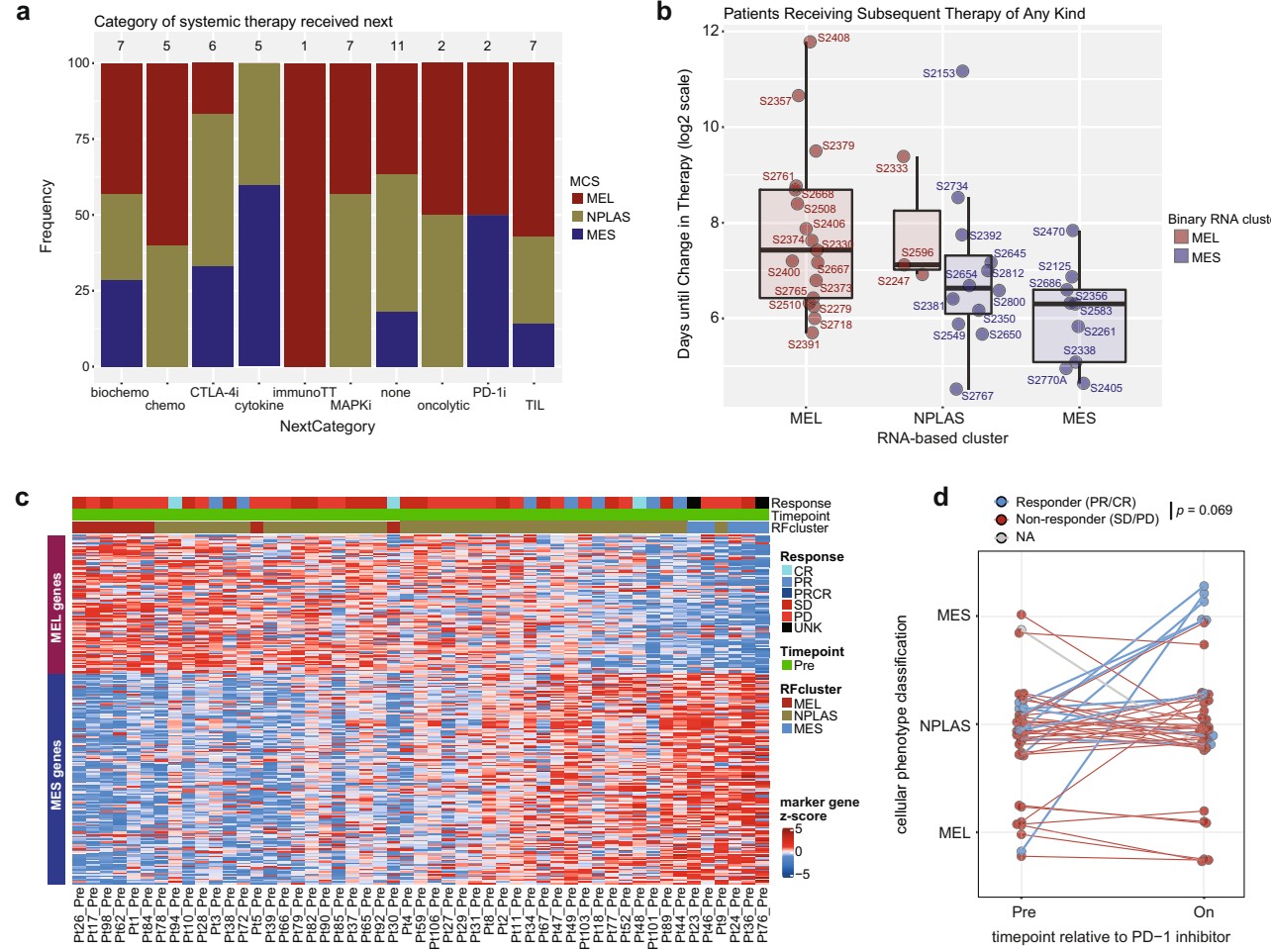

**Fig. 7 Clinical associations of MCS. a** Systemic therapies received by patients (*n* = 53) subsequent to the tumor harvests from which the early passage cell lines were generated, grouped by treatment category. The total number of patients in each category is indicated above each bar, and the MCS of each tumor-derived cell line is indicated in color. **b** Time, in days, from the initiation of the next line of systemic therapy after cell line-generating tumor harvest, until a further change in therapy was required or last follow-up. Boxplots indicate the median (thick bar), first and third quartiles (lower and upper bounds of the box, respectively), lowest and highest data value within 1.5 times the interquartile range (lower and upper bounds of the whisker), and all individual data points (*n* = 41) are shown. **c** Heatmap of MEL and MES gene expression in pre-PD-1 inhibitor treatment samples of the Riaz cohort, ordered by decreasing MEL-MES score (see "Methods"). Objective response to treatment (Response), sampling time point (Time point; all pre) and random forest model-assigned cluster (RFcluster) are shown for each sample. **d** Longitudinally sampled patients within the Riaz PD-1 inhibitor cohort revealed a trend towards higher proportion of responders in patients demonstrating a MES-wise shift (i.e. MEL to NPLAS, MEL to MES, or NPLAS to MES) in model-assigned cluster type (*p* = 0.069, two-sided Fisher exact test). Source data are provided as a Source Data file.

## Discussion

We hypothesized that unraveling the multiple molecular inputs governing MCS would reveal previously unrecognized biological interactions that drive clinically relevant outcomes. We worked with early passage patient-derived melanoma cell lines for two reasons: (1) to overcome the challenge of tumor heterogeneity, and (2) to explore multi-omics that are not currently available by single-cell methods. In this way, we determined that MCS, representing distinct biological capabilities, are highly conserved in patient tissue samples and are differentially associated with immune gene profiles and immune cell content within the tumor microenvironment in vivo. Individual MEL or MES genes that

displayed similar expression patterns (e.g. switch-like, graded, etc.), were subject to distinct combinations of molecular regulation at genomic, transcriptional and post-transcriptional levels. Multiplatform analyses revealed a high degree of regulatory heterogeneity even within each MCS.

Most notably, we found that the simultaneous action of multiple modalities of gene regulation provides differential opportunity for cellular plasticity; specifically, the MEL state was associated with less readily reversible regulatory modalities (e.g. CNA), suggesting that it may be more molecularly 'fixed' or 'stable'. The relative minority of MES samples in our own patient cohort, as well as from high-purity samples of the TCGA melanoma cohort, may thus reflect a lower frequency of cancer cells that are able to escape the MEL state to become the dominant population in melanoma tumors. Predominance of the MEL state was also observed in a single-cell RNA-Seq analysis of melanoma tumors in which 7 of 10 tumors at bulk level displayed an *MITF*-high (here MEL) phenotype, despite all tumors evaluable at the single-cell level ($n = 7$) showing admixed *MITF*-high and *AXL*-high (here MEL and MES, respectively) phenotype cells[6]. Depending on prevailing microenvironmental pressures, in vivo studies also suggest that the MES state may inherently tend towards reversion to a MEL state[68]. These data are interesting in light of the lower frequency of MES (or analogous *MITF*-low/*AXL*-high) samples across many published melanoma cell line and patient tumor cohorts[1,64], indicating that the higher prevalence of the MEL state in cell line panels is unlikely to be explained solely by a selective advantage to sustained in vitro culture.

Examination of the specific underlying regulatory modalities unexpectedly revealed that samples which clustered in the non-MEL/non-MES group (here called NPLAS) did not simply represent a mixture or a transitional state between MEL and MES and may in fact be a distinct melanoma cell state. Other recent reports have found additional support for multiple melanoma cell states and a molecular basis for variable stability of such states[69,70]. Indeed, as multi-dimensional molecular profiling becomes more feasible, the number of distinct cell states detected within each tumor may continue to increase; further functional analyses will need to be combined with 'omics' studies in order to determine the number of states beyond which further subdivision is no longer clinically relevant. However, as therapeutics evolve, so too will the meaningful groupings of melanoma cell states. Overall, it is clear that gene expression profiling alone will not be sufficient to identify high-yield molecular therapeutic targets. By identifying multiple regulatory features that simultaneously underlie MCS, our study suggests many potential combinatorial treatment strategies aimed at different cell states within a tumor.

In our analyses, all cohorts of clinical samples demonstrated clear evidence of segregation into MCS, which were in turn associated with distinct immune microenvironments in tumors, including immunosuppressive populations in MES samples. Interestingly, we observed an apparent shift in the MCS within tumors toward a more MES state in longitudinal samples responding to PD-1 CPI, suggesting depletion of more therapeutically susceptible MEL cell content in response to PD-1 blockade. These results are consistent with the expression of melanocyte differentiation antigens in MEL cells, which are highly immunogenic and targets of anti-cancer CD8+ T cell clones. Conversely, isolated MES cells intrinsically express *CCL2* and *PDCD1LG2*, which can facilitate a pro-tumor microenvironment by attracting monocytes and inhibiting T-cell proliferation/cytokine production, respectively[71,72]. Several MEL genes (*IRF4*, *IL16*, *IL6R*) that are known to be associated with an anti-tumor milieu (e.g. lymphocyte activation, chemotaxis, and differentiation)[73] appear actively suppressed in MES cells through mechanisms such

as miR-125b-5p targeting and DNA methylation, reinforcing a potentially immunotherapy-resistant microenvironment.

These data confirm the utility of performing parallel 'omics' analyses to identify multi-modal molecular regulatory features and relating these to observable clinical outcomes. The determinants of MCS cannot be adequately inferred from a single data type, are complex, and may cumulatively determine the degree of cellular plasticity. We provide additional data supporting the molecular independence of the NPLAS state, still largely thought to represent a transition or mixed state between MEL and MES extremes. Importantly, the large cohort of patient-derived cell lines will serve as an ongoing resource to the research community. The molecular characterization we performed using five distinct 'omics' platforms can be further integrated with data from emerging molecular techniques and functional studies to guide future strategies targeting melanoma cell-intrinsic phenotypes as independent or adjunct anti-cancer therapies.

## Methods

**Ethics Statement.** The University of Texas MD Anderson Cancer Center Institutional Review Board (IRB) approved our study involving melanoma cell line generation from metastatic tumor specimens as part of the Adoptive T-cell Therapy Clinical Program (LAB06-0755 and 2004-0069). Written informed consents were provided by the participants prior to enrollment, including consent to harvest tumor material and use in parallel cancer-related research as presented in this study. All experimental methods abided by the Declaration of Helsinki. Participants did not receive compensation for their participation.

**Cell Lines.** Melanoma cell lines (referred to here as "MDACC cell lines") were generated previously[74,75], as follows. Briefly, each specimen from a metastatic melanoma tumor was collected and processed into a single-cell suspension by incubation with an enzymatic digestion cocktail (0.375% collagenase type I, 75 μg/mL hyaluronidase, and 250 U/mL DNase-I) in tumor digestion medium (RPMI-1640 containing 10 mM HEPES, 1% penicillin/streptomycin and 20 μg/mL gentamicin; Gibco/Invitrogen) in a humidified incubator at 37 °C with 5% $CO_2$ with gentle rotation for 2–3 h. The tumor digest was filtered through a 70 μm filter, washed in sterile PBS, and re-suspended in serum-free media, which was then plated in one well of a 6-well culture plate and incubated at 37 °C in a 5% $CO_2$ atmosphere. After 24 h, the media was replaced with fresh complete tumor media, comprised of RPMI-1640 supplemented with 1% GlutaMAX, 10% FBS, 1% penicillin/streptomycin, 20 μg/mL gentamicin, 50 μM β-mercaptoethanol (Gibco/Invitrogen), 10 mM HEPES, and 5 μg/mL insulin-selenium-transferrin (Gibco/Invitrogen). Cells were grown-on in enriched DMEM/F12 culture media (Gibco/Invitrogen) supplemented as above and with 1 mM sodium pyruvate and MycoZap-PR (Lonza). Once enough cells were grown in pre-cell line culture, tumor cell purity was tested using a melanoma tumor surface marker (MCSP-1) by flow cytometry. Controlled serum starvation was performed to eliminate fibroblasts. Cultures were deemed established when the cells stained positive for a melanoma tumor marker (MCSP, 1:50, Miltenyi, cat# 130-117-347) and negative for a fibroblast marker (CD90, 1:50, BD Pharmingen, cat# 561558). All the cell lines were derived from metastatic melanoma tumors that had advanced to various sites (e.g., lymph nodes, soft tissues, lungs). Tissues were harvested primarily by surgical resection and cell lines were generated by the MD Anderson Cancer Center TIL Laboratory between 2007 and 2014. All MDACC cell lines were used within 10 passages of tumor line establishment; cells were cryopreserved and kept in stocks in liquid nitrogen until use.

Commercially available human melanoma cell lines MEL888, SKMEL23, WM902B, and WM115 were generously provided by Dr. Michael A. Davies at the University of Texas, MD Anderson Cancer Center.

All cell lines were tested for mycoplasma using the MycoAlert detection kit (Lonza) and authenticated by STR fingerprinting compared with matched-patient peripheral blood mononuclear cell samples to confirm identity.

**Nucleic acid extraction.** Cell line-derived DNA was extracted from freshly isolated cell pellets using the DNeasy Blood and Tissue kit (Qiagen) and eluted in TE buffer. DNA concentration was measured using a Quant-iT PicoGreen assay kit (ThermoFisher) and quality assessed using an Agilent Bioanalyzer 2100 and DNA 1000 kit (Agilent). A total of 250 ng DNA was submitted for DNA sequencing. A total of 1 μg DNA was submitted for DNA methylation analysis.

Total RNA including miR were extracted using the mirVana microRNA Isolation Kit (ThermoFisher) and eluted in nuclease-free water. RNA was quantitated using a Qubit fluorimeter and quality was assessed using a Fragment Analyzer (Advanced Analytical). Only analytes with an RNA Quality Number (RQN) ≥ 7.0 proceeded to RNA sequencing. For miR (small RNA) sequencing, total RNA quality was assessed using the Bioanalyzer 2100 and RNA6000 Nano

assay (Agilent); all analytes had RIN ≥ 8.0, except for three cell lines; 2350 (RIN 7.9), 2417 (RIN 6.9) and 2734 (RIN 5.2). All samples proceeded to miR sequencing given that RIN values may be poorly reflective of sample miR quality[76].

**RNA sequencing library construction and sequencing**. Illumina compatible libraries were prepared using the TruSeq Stranded Total RNA LT Sample Prep Kit with Ribo-Zero Gold (Illumina, Inc.), per the manufacturer's protocol. Briefly, 250-1000 ng of total RNA was DNase-I treated and then depleted of ribosomal RNA (rRNA) using biotinylated target-specific oligos. Following purification, the RNA was fragmented using divalent cations and first-strand cDNA synthesis carried out using random primers. Second-strand cDNA synthesis was then performed, and the ends of the resulting double-stranded cDNA fragments were repaired, 5′-phosphorylated and 3′-A tailed. Illumina-specific Y-shaped indexed adapters were then ligated. The products were purified and enriched with 12 cycles of PCR to create the final cDNA library. The libraries were quantified fluorometrically using the Qubit™ dsDNA HS Assay (ThermoFisher) and assessed for size distribution using the Fragment Analyzer (Advanced Analytical) or a TapeStation 2200 (Agilent), then normalized and pooled, 20 or 24 samples per pool. The pools were sequenced in 3 and 4 Illumina HiSeq3000 lanes respectively, using the 75 bp paired-end format. Following sequencing BCL files were converted to FASTQ files and individual sample libraries were de-multiplexed and adapters trimmed using bcl2fastq2 Conversion Software (v2.17.1.14). RNA reads were aligned to the GRCh37/hg19 genome assembly using STAR (v2.3.0)[77]. Gene expression was quantified using htseq-count[78] against a customized annotation compilation based on UCSC known-gene annotations (v.June 2011). Log2-transformed transcripts per million (TPM) values were used for subsequent analysis. For cross-cohort analyses, raw read counts were normalized using the variance-stabilizing transformation implemented in R package DESeq2 v1.32.0[79].

**Transcriptome analysis**
*Unsupervised clustering*. For clustering, a set of 1500 protein-coding genes that were most variably expressed (highest median absolute deviation, MAD) across 68 melanoma cell line samples were identified. Consensus negative matrix factorization (cNMF) clustering was implemented using the GenePattern NMFConsensus module (Broad Institute) with 2- to 5-cluster-solutions and 1,000 iterations[80]. Silhouette plots and cophenetic correlations were assessed to define the number of clusters in the optimal cluster solution ($k = 3$), defining three subtypes: cluster 1 ($n = 27$), cluster 2 ($n = 22$), and cluster 3 ($n = 19$) (Supplementary Fig. 1a–c). A similar cNMF approach was applied to data of other platforms: miR (input = all 623 expressed mature strand miRs), protein (input = 287 proteins/modifications), and DNA methylation (input = top 2,500 most variable probes) (Fig. 3b). NPLAS cell lines from the MDACC cohort were defined as more MEL-like or more MES-like based on their grouping with MEL or MES samples, respectively, following unsupervised hierarchical clustering of the refined cell line cohort ($n = 53$) using MEL/MES gene expression, Euclidean distance and complete linkage, cutting the dendrogram at $k = 2$.

*Gene set analyses*. Gene set enrichment analyses were performed using the Broad Institute's GSEA desktop module v4.1.0 with default settings[81]. Cluster membership comparisons between datasets were performed using the SubMap algorithm using the original code as described by Hoshida et al.[39], dated 14 Oct 2008 downloaded from the Broad Institute GenePattern module archive on 29 July 2021. R package biomaRt (v2.48.2) was used where necessary to cross-map gene identifiers in the datasets analyzed.

*Cluster-defining gene set identification*. After identifying the transcriptome-defined sample clusters as above, we identified the statistically significantly differentially expressed genes between clusters using pairwise comparisons between clusters with a cutoff of unpaired t-test FDR < 0.05 and 10 times fold change. A total of 98 significantly upregulated and 149 significantly downregulated genes were identified in the MDACC cell line cluster 1 versus cluster 3 samples. Inspection of these cluster-defining genes with regard to known biological functions of included genes permitted the classification of a melanocytic-like cluster (cluster 1, MEL), a mesenchymal-like cluster (cluster 3, MES) and a neural-plastic cluster (cluster 2, NPLAS). Functional annotation of these MEL and MES gene sets within Gene Ontology – Biological Process (GO-BP) annotations and also within KEGG pathways was performed using DAVID (v6.8, http://David.ncifcrf.gov)[82,83].

*Long non-coding RNA (lncRNA) analysis*. RNA-Seq reads were aligned to GRCh37/hg19 (Ensembl 74) and annotated with GENCODE (release 19), and the expression of long non-coding RNAs (lncRNA) was determined as transcripts per million (TPM) using the RSEM algorithm (v1.2.12). After filtering out the lncRNAs having TPM values < 1, a total of 586 lncRNAs remained. For the $n = 53$ melanoma MDACC cell line cohort, we used SAM two-class unpaired analyses to identify lncRNA that were differentially expressed between MCS: MEL ($n = 21$) and MES ($n = 11$), MES and other ($n = 42$), MEL and other, and NPLAS ($n = 21$) and other ($n = 32$) (R package samr v2.0, with settings nperms = 1000, testStatistic = "wilcoxon", center.arrays = FALSE, and fdr.output = 0.05). For each SAM run the input was a matrix of normalized TPM abundance, in which we required lncRNAs

to have a mean TPM ≥ 1.0 in either group being compared, and |fold change (FC)| ≥1.25. This filtering resulted in inputs of 592 lncRNAs for the MEL vs MES comparison, 567 lncRNAs for MES vs other, 445 lncRNAs for MEL vs other, and 398 lncRNAs for NPLAS vs other. In SAM outputs we retained only lncRNAs with q-value < 0.05. We generated summary barplots showing the 20 largest positive and negative FC ≥2.0, clipping FCs at ± 40, and displaying the FC, then the mean TPM in each group to the right of each barplot (Supplementary Fig. 8).

**MicroRNA sequencing**. For each sample, 1 μg of total RNA was used to prepare a small RNA library using the TruSeq Small RNA Library Prep Kit (Illumina), according to the manufacturer's recommended protocol. Libraries were purified using BluePippin DNA size selection in the range 125–160 bp (Sage Science) and the quality was assessed using the Bioanalyzer DNA HS Assay (Agilent) to confirm appropriate fragment size, complexity, and absence of adaptor dimers. These purified miR libraries were then sequenced using 35 bp single-end reads on an Illumina HiSeq 2500 System.

miR sequencing reads were aligned to the human mature miR sequences of miRBase release 20, and numbers of reads with an exact-match alignment were calculated. From 2588 miRs that were quantified per sample, those with no count in all the samples were removed, resulting in a total of 2196 expressed miRs. Mature strand miR counts were quantified by a customized pipeline which used bowtie2 to search for the longest exact match (parameters: -norc -N 1 -L 16 -k 5 -local): Of the 2196 expressed miRs, 623 were expressed in at least 90% of all samples, and were used in unsupervised clustering and subtype marker identification. We further identified miRs upregulated in RNA-defined MCS, using a 10 times fold change threshold between MEL vs MES with an unpaired t-test FDR < 0.05. MiR sequencing data were not available for cell lines S2365, S2391, and S2654 (refined cohort) and S2316, S2354, S2399, and S2844 (complete cohort).

*Determining miR targets within MEL/MES marker genes*. To identify the MEL or MES genes targeted by miRs we used three computationally derived target databases and two experimental databases as follows.

Computationally derived databases: miRDB v5.0[84] (http://www.mirdb.org/download.html); TargetMiner 2012 release[85] (https://www.isical.ac.in/~bioinfo_miu/download20.htm); TargetScan v7.1[86] (http://www.targetscan.org/vert_71/). Experimentally-derived/validated databases: miRTarBase v6.1[87,88] (http://mirtarbase.mbc.nctu.edu.tw/php/index.php); miRWalk v2.0[89,90] (http://zmf.umm.uni-heidelberg.de/apps/zmf/mirwalk2/).

We considered a gene to be a miR target if the miR-gene pair was found in at least one experimental or two computational databases. Anti-correlation with mRNA in the cell lines was then combined with these verified miR-gene pairs to identify putative regulatory relationships of interest (Pearson's correlation coefficient, FDR ≤ 0.1).

**Whole-Exome Sequencing**. Whole-exome sequencing (WES) of 250 ng of DNA from melanoma cell lines and matched normal blood samples was performed using the Agilent SureSelect Human All Exon 44 Mb v2.0 bait set (Agilent Technologies, USA)[91,92]. Briefly, genomic DNA was sheared, end repaired, ligated with barcoded Illumina sequencing adapters, amplified, size selected, and subjected to in-solution hybrid capture. The resulting exome Illumina sequencing libraries were then qPCR quantified, pooled, and sequenced with 76 base paired-end reads using Illumina GAII or HiSeq 2000 sequencers (Illumina, USA). BCL files were processed using Illumina's Consensus Assessment of Sequence and Variation (CASAVA) tool (Illumina, USA; https://www.illumina.com/documents/products/datasheets/datasheet_genomic_sequence.pdf). Read alignment and processing were performed using BWA-aln[93] and Picard (http://broadinstitute.github.io/picard/), to genome assembly GRCh37/hg19. Tumor coverage was up to 150×, with a median of 122 million read pairs generated per sample. WES data were not available for cell lines S2350 and S2667 (refined cohort), and S2844 (complete cohort).

*Somatic variant calling and significance analysis*. Somatic mutations and short insertions/deletions (indels) were called and post-filtered using the MuTect2 module of GATK (v3.8.1.0) with default settings[94,95]. Variants that did not pass default alt_allele_in_normal or germline risk thresholds were excluded. Identified variants were then annotated to genes, transcripts, and variant severity using Annovar rev 521 (2013 May)[96]. The variant allele frequency threshold was set at 5%.

*Somatic copy-number alteration and purity analysis*. Somatic copy-number alterations (CNA) in melanoma cell lines were identified from WES data using ExomeCNV[97] (ExomeLyzer v1.6.2) and HMMCopy v1.24.0[98]. Segmented data was processed using GISTIC2 (v2.0.21), and values in focal_data_by_gene.txt were used to define copy-number alterations. Genes having log2-copy-number value < −0.5 were considered to have losses, and those with values = 0.5 were considered to have amplifications. Cell line purity and ploidy were estimated from WES data using Sequenza v2.1.0 (2014 Oct)[99]. Direct correlation between CNA and transcript expression was assessed for each MEL and MES gene using one-sided Spearman correlation tests with $p < 0.05$ considered significant.

**DNA methylation analysis**. DNA methylation was evaluated using the Illumina Infinium Human Methylation 450 K BeadChips (HM450K; Illumina)[100]. Briefly, 1 μg of genomic DNA was bisulfite converted using the EZ DNA Methylation-Direct kit (Zymo Research Irvine, CA). The bisulfite conversion efficacy was evaluated using the MethyLight assay for a panel of defined markers. Samples passing quality control (QC) were whole genome amplified, enzymatically fragmented, hybridized onto HM450K BeadChips, and scanned using the Illumina iScan microarray scanner (Illumina). The raw data (in IDAT file format) was processed with the 'lumi' R package using default parameters, and DNA methylation values were calculated as M values for use in subsequent analyses. Annotation of HumanMethylation450 probes was downloaded from the Illumina website ('https://support.illumina.com/downloads/humanmethylation450_15017482_v1-2_product_files.html'). Probes associated with individual genes were corrected according to gene location information from GENCODE Human v19. Briefly, probes located within 2000 bp upstream of the gene start site were annotated as 'TSS2000' or 'Promoter', and probes located between gene start and gene end were annotated as 'GeneBody'. Existing annotations of 5'UTR or 3'UTR were retained. DNA methylation data were not available for two cell lines in the refined cohort (S2350, S2667).

**Reverse-phase protein array (RPPA)**. Samples were analyzed at the RPPA Core Facility at The University of Texas MD Anderson Cancer Center. Protein was extracted from PBS-washed cell pellets using RPPA lysis buffer (1% Triton X-100, 50 mmol/L HEPES (pH 7.4), 150 mmol/L NaCl, 1.5 mmol/L MgCl$_2$, 1 mmol/L EGTA, 100 mmol/L NaF, 10 mmol/L NaPP$_i$, 10% glycerol, 1 mmol/L phenylmethylsulfonyl fluoride, 1 mmol/L Na$_3$VO$_4$, and aprotinin 10 μg/mL). Cell lysates were adjusted to 1.5 μg/μL concentration after estimation by the bicinchoninic acid assay (BCA) and boiled with 1% SDS without bromophenol blue for 5 min. Each sample had three biological replicates.

Cell lysates underwent five 2-fold serial dilutions in lysis buffer (from undiluted to 1:16 dilution) and were printed on nitrocellulose-coated slides (Grace Bio-Labs, Inc., Bend, OR) by Aushon Biosystems 2470 Arrayer (Quanterix Corp., Lexington, MA). Slides were probed with a total of 291 primary antibodies (validated by the RPPA Core) followed by corresponding secondary antibodies (biotin-conjugated goat anti-rabbit IgG, goat anti-mouse IgG or rabbit anti-goat IgG). Signal was captured using the CSA amplification approach and visualized by 3'-diaminobenzidine (DAB) colorimetric reaction. Slides were scanned in a CanoScan 9000 F scanner to produce 16-bit tiff images. Spots from tiff images were identified and the densities quantified using Array-Pro Analyzer (Meda Cybernetics) to generate signal intensities, then processed using SuperCurve[101] to estimate the relative protein expression levels, and were normalized by median polishing. During this process, raw spot intensity data were adjusted to correct for spatial bias before model fitting. A QC metric[102] (range, 0-1) was returned for each slide, and only those slides with QC score above 0.8 were retained for further analysis. If more than one slide was stained with an antibody, the slide with the highest QC score was used for analysis. The final selection of antibodies was determined by the availability of high-quality antibodies that consistently passed a strict validation process, including high correlation ($R \geq 0.7$) with single-probe immunoblots, absence of non-specific binding, proportional change in signal following perturbation of the target (phospho)protein, and examination of protein-mRNA correlation[103]. In total, 291 antibodies (181 "validated", 101 "use with caution", 4 "under evaluation", 5 "QC") were evaluated for 67 of the 68 cell lines studied herein in triplicate. RPPA data were not available for cell line S2844. Antibody information (company, catalog number, dilution, species) and normalized log2 values for each protein within each sample are provided in Supplementary Data 5.

**Integrative analysis**. For the combined assessment of molecular regulatory modalities, CNA, methylation and miR acting upon each MEL or MES gene (Supplementary Data 2) were expressed in the form of directed binarized tables, wherein −1 indicated significant inverse correlation, +1 indicated significant direct correlation, and 0 indicated no significant correlation with gene transcript abundance, with statistical significance taken as per the thresholds defined for each individual data type. Individual methylation probes displayed either direct or inverse correlations with host gene transcript abundance; in cases where *equal* numbers of methylation probes were statistically significantly correlated in both directions (details for each gene provided in Supplementary Data 6), the consensus value used for regulatory modality clustering was taken as the direction of the probe with the most significant q-value. To identify gene relationships based on these regulatory modalities, each of the MEL and MES genes was subjected to unsupervised clustering after generating dissimilarity matrices based on a generalization of Gower's formula as implemented by the "daisy" function of the "cluster" package (v2.0.7-1) in R, due to the inclusion of categorical input variables. The resulting dendrograms were then plotted radially and annotated for each type of molecular regulation for each gene (Fig. 6a, b).

**Public transcriptome datasets**

*TCGA melanoma dataset*. Normalized gene transcript abundances of 471 melanoma tumor samples contained within the TCGA melanoma ("SKCM") cohort[42] were downloaded from http://gdac.broadinstitute.org/ using the Broad Firehose

2016 version. Tumor purity information was obtained from[42,104]. The samples were filtered by selecting only metastatic tumors ($n = 368$) and further by high tumor purity (≥0.85) determined by ABSOLUTE and/or CPE[104,105], resulting in a total of 77 samples for "high-purity metastatic" sample analyses and 368 samples for "any-purity metastatic" sample analyses.

*Cancer cell line encyclopedia (CCLE)*. Gene expression (TPM) and segmented copy-number profile data (segmeans) and annotations for all CCLE cell lines were downloaded via the CCLE portal at https://portals.broadinstitute.org/ccle (accessed October 18, 2020). Due to evolution of the CCLE dataset over time, cell lines annotated as melanomas (Histology = "malignant_melanoma", type_refined = " melanoma", tcga_code = "SKCM") were then matched between gene expression and copy-number datasets to identify only samples with both types of data and confirmed as melanoma lines by manual review ($n = 60$). MCS of each cell line was determined by unsupervised cNMF clustering based on the expression of the MEL/MES genes using $k = 3$ clusters (Supplementary Fig. 6a). Segment mean data were converted to absolute CN estimates using the formula CN = (2^segmean)*2 and CNA frequency plots generated using the "cnFreq" command in the R package GenVisR v1.14.2[106] applied to samples of each MCS using default thresholds for considering segment gains (CN_high_cutoff = 2.5) or losses (CN_low_cutoff = 1.5) (Supplementary Fig. 6b).

*Immune checkpoint inhibitor-treated clinical datasets and response definitions*. Transcriptome data (raw counts) from a PD-1 inhibitor-treated melanoma clinical and gene expression dataset (Riaz et al.)[67] were downloaded from https://github.com/riazn/bms038_analysis/tree/master/ (accessed 10/17/20), and response definitions were utilized as provided in the paper, indicated as radiographic objective response categories according to standard RECIST v1.1 criteria including subcategorization into complete response (CR), partial response (PR), stable disease (SD), or progressive disease (PD).

**Transcriptomic machine learning classification of melanoma intrinsic programs**. To train models for the classification of individual samples into one of the three MCS based on the expression of the MEL and MES genes, the caret package (v6.0-81) in R was used to generate a random forest (RF) classifier model taking cNMF-assigned cluster membership as the class labels for training. In keeping with a melanoma cell-intrinsic phenotype, the RF model was trained on the MDACC cell line cohort which due to limited sample size was not amenable to use for randomized split training and test subsets. Input data was normalized gene expression data using the variance-stabilized transformation applied to raw count input as implemented with default settings using DESeq2 (v1.32.0) in R, in order to maximize equivalence to vst-transformed data of other clinical datasets for which raw count data tables are commonly available. The MDACC cell line cohort MEL and MES gene set expression data was pre-processed using the preProcess() function with method=c("nzv","corr"), to retain non-zero value features and remove highly correlated features; a random seed was set to 10110 and the RF model trained using the train() function with method = "ranger", importance = "impurity", metric = "Accuracy", tuneGrid=expand.grid(mtry = c(1:50), splitrule=c("gini","extratrees"), min.node.size = 1) and trControl=trainControl(method = "repeatedcv", number = 10, repeats = 10, selectionFunction = "tolerance", sampling = "down"). The final values used for the model were mtry = 1, splitrule=extratrees, and min.node.size = 1 with an overall accuracy of 0.97 [95%CI 0.898–0.996], $p = 2.2e16$, kappa = 0.96 based on cross-validation.

**Marker gene analyses and immune deconvolution**. For each sample, a MEL-MES score was calculated as the difference between the averaged expression of MEL and MES marker genes, i.e. (MEL_average–MES_average).

Immune, stromal and tumor composition of clinical samples was inferred from transcriptomic data (TPM or VST-normalized counts) using MCP-counter (v1.2.0)[43] which enumerates several broad immune cell lineages as well as fibroblasts and endothelial cells. Additional deconvolution methods (EPIC, quanTIseq, TIMER) were implemented using R package "immunedeconv" v2.0.4[107]. For individual marker gene analyses, normalized expression data (TPM or VST-normalized counts) were used. A cytolytic gene signature was calculated as previously, using the geometric mean of normalized expression data for *GZMA* and *PRF1*[108]. An M2-macrophage gene signature was developed similarly, using the geometric mean of gene expression values for *MRC1*, *CD163*, and *IL10*.

**MiR-211-5p targeting of ZEB1**. Cell lines as indicated in the results were treated with miR-211-5p mimic at 20 nM (Cat. No. 4464066, ThermoFisher Scientific, MA, USA) or scrambled control (Cat. No. 4464058, ThermoFisher Scientific, MA, USA) using lipofectamine RNAiMAX transfection reagent (Cat. No. 1378150, ThermoFisher Scientific, MA, USA) according to the manufacturer's recommendations, and the effect on ZEB1 mRNA level evaluated by RT-PCR using ZEB1 TaqMan gene expression assays (Assay ID: Hs00232783_m1, Cat. No. 4331182, ThermoFisher Scientific, MA, USA), and GAPDH TaqMan gene expression assay (Assay ID: Hs99999905_m1, Cat. No. 4331182, ThermoFisher Scientific, MA, USA) as control, using the ABI StepOnePlus real-time PCR-system (Cat. No. 4376600, ThermoFisher Scientific, MA, USA). All quantitative

PCR (qPCR) assays were conducted with a total reaction volume of 20 μL, according to the manufacturer's instructions. ZEB1 protein level was measured by Western Blot using the rabbit anti-human ZEB1 (H-102) polyclonal IgG (1:500, Cat. No. sc-25388, Santa Cruz Biotechnology, CA, USA), and mouse anti-GAPDH monoclonal IgM (1:10000, clone GAPDH-71.1, Cat. No. G9295, Sigma Aldrich, MO, USA) as house-keeping control.

**Cell viability assays with BRAF inhibitor or decitabine**. Cell viability was determined using the MTT assay kit (Cat 11465007001, Sigma-Aldrich) according to the manufacturer's instructions. Briefly, $5 \times 10^4$ cells were seeded per well in 100 μL DMEM (HyClone; Thermo Scientific, 90 Logan, UT) supplemented with 10% fetal bovine serum (FBS) (Atlanta Biologicals, 91 Lawrenceville, GA) and 1% penicillin/streptomycin mixture (Sigma-Aldrich, MO). Melanoma cell lines treated with decitabine were considered MEL-like or MES-like based on the protein expression of E-cadherin versus N-cadherin and ZEB1, respectively, which showed similar results to our patient-derived cell lines. For the experiments, cells were exposed to increasing concentrations of either the BRAF inhibitor vemurafenib (Roche, IN) for 24 h or the demethylating agent decitabine (Sigma-Aldrich, MO) for 72 h, or DMSO vehicle control. At the end of treatment, the cells were incubated with 10 μL/well of the MTT labeling reagent for 4 h, then incubated with 100 μL of the Solubilization solution overnight. Absorbance of each sample was measured on a microplate reader (Molecular Devices, CA) at the wavelength of 560 nm.

**General statistical considerations**. Data were collected/collated directly in tabulated form using Microsoft Excel (Excel 2016 or later versions) or R (v4.1.0). Because of the nature of the current study utilizing a very limited patient-derived resource (generation of melanoma cell lines from metastatic tumor specimens as part of IRB-approved Adoptive T-cell Therapy Clinical Program), the sample size was not calculated beforehand. The total number of 68 patient-derived, early passage melanoma cell lines and the 53 in the refined cohort are arguably the largest of their type analyzed using multiple platforms to this extent, and we believe that our data, made publicly available here, will serve the research community to further the understanding of this disease. The University of Texas MD Anderson Cancer Center has a large and active melanoma service and TIL therapy program which attracts patients from within and beyond the local geographic catchment, thus the recruited patient population is expected to be representative of the broader advanced melanoma patient population. Additionally, not all patients had TIL harvest procedures performed at the same stage of their disease, thus minimizing biases introduced by sampling only patients at a time point of very advanced and heavily pre-treated disease. For inclusion in this study patient samples did have to produce a cell line suitable for analysis however as the success of this process is somewhat random it is not subject to any known/proven systematic sources of bias. Between-group comparisons of genomic data utilized the Mann–Whitney $U$ test (two groups) or Kruskal–Wallis (>two groups) with post-hoc Dunn tests when appropriate, and for categorical variables/features the Fisher's Exact test (two-tailed). Comparisons were generally considered statistically significant at $p < 0.05$ unless otherwise indicated, including multiple comparison correction, where appropriate, at a false-discovery rate threshold of 0.05. For box-whisker plots, the box represents the first (lower bound) and third (upper bound) quartiles, with median bar, and the whiskers indicate the most extreme data value within 1.5 times the interquartile range above or below the third or first quartiles, respectively.

**Reporting summary**. Further information on research design is available in the Nature Research Reporting Summary linked to this article.

## Data availability

The genomic data generated in this study have been deposited in the European Genome-Phenome Archive (EGA) under accession EGAS00001004536. These data are available upon request to the corresponding author for academic cancer research purposes in accordance with the conditions of consent agreed to by the source participants. Requests will be addressed within 8 weeks and, if approved, access will be made available for a one-year period, renewable upon additional request. Relevant non-identifiable clinical metadata and processed data generated in this study are provided in the Supplementary Data and Source Data. The Cancer Genome Atlas (TCGA) melanoma ("SKCM") cohort publicly available gene expression data used in this study are available from http://gdac.broadinstitute.org/ as the "Broad Firehose 2016" version. Publicly available Cancer Cell Line Encyclopedia (CCLE) gene expression (TPM), segmented copy-number profile data (segmeans) and annotations are available via the CCLE portal at https://portals.broadinstitute.org/ccle (registration required). Publicly available transcriptome data (raw counts) from the PD-1 inhibitor-treated melanoma clinical and gene expression dataset of Riaz et al.[67] are available from https://github.com/riazn/bms038_analysis/tree/master/. Source data are provided with this paper. The remaining data are available within the Article, Supplementary Information or Source Data file. Source data are provided with this paper.

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

## Acknowledgements

This study was supported through a National Health and Medical Research Council of Australia CJ Martin Early Career Fellowship (MCA, #1148680); a Developmental Research Project Award, National Institutes of Health/NCI Cancer Center Support Grant (No. P30 CA016672; S.E.W. and C.B.); NIH R50 Grant (#R50CA221675; Y.L.); The University of Texas MD Anderson Cancer Center Melanoma Moon Shot Program (S.E.W., C.B., A.J.L., J.A.W., J.E.G., M.A.D.); a Faculty Award (SEW); a Rising Stars Award, The University of Texas System (S.E.W.); The University of Texas MD Anderson Cancer Center NCI SPORE in Melanoma (S.E.W., P50 CA093459); Dr Miriam and Sheldon G. Adelson Medical Research Foundation (D.S.B.H., E.A.G., P.H., M.A.D.); AIM at Melanoma Foundation (E.A.G., M.A.D.); the Miriam and Jim Mulva Foundation (E.A.G.); a Cancer Prevention and Research Institute of Texas (CPRIT) Core Facilities Support Award (J.S., CPRIT RP170002); The University of Texas MD Anderson Cancer Center Support grant (NIH/NCI P30 CA016672),; generous philanthropic contributions to The University of Texas MD Anderson Cancer Center Melanoma Moon Shots Program from the Lyda Hill Foundation; and utilized platform assistance from the Cancer Genomics Laboratory and Immunotherapy Platform. Additional support was provided from the Cancer Prevention Research Institute of Texas (R1205 01 to P.A.F.) and Welch Foundation (G-0040 to P.A.F.). MAD receives research support from the National Institutes of Health and National Cancer Institute (NIH/NCI)(1 P50 CA221703-02), the American Cancer Society and the Melanoma Research Alliance, Cancer Fighters of Houston, the Anne and John Mendelsohn Chair for Cancer Research. J.E.G. also receives support from the Robert and Lynne Grossman Family Foundation and the Michael and Patricia Booker Melanoma Research Endowment.

## Author contributions

Conceptualization: S.E.W., P.A.F., C.B., M.A.D. Investigation: M.C.A., C.W., T.K., J.O., R.S.S., C.A.C., M.A.F., X.Y., H.Z., X.S., X.M., D.S.B.H., Y.L., A.G.R., K.M.W., A.J.L., M.A.B., J.S., Y.C., K.P.P., G.R., P.L. Provision/Acquisition of Data and Materials: E.M.B., L.E.H., M.A.D., C.B., S.E.W., J.A.W., K.R., L.N.K. Formal Analysis: M.C.A., C.W., T.K., A.G.R., J.O., S.E.W. Data Curation: M.C.A., C.W., J.O., L.E.H. Visualization: M.C.A., C.W., A.G.R., J.O. Supervision: S.E.W., C.B., P.A.F., J.Z. Funding Acquisition: M.B.M., C.L.H., D.S.B.H., E.A.G., P.H., J.Z., J.E.G., M.A.D., P.A.F., C.B., S.E.W. Writing—Drafting & Editing: M.C.A., J.O., S.E.W. Writing—Review and Approval of Final Manuscript: all authors

## Competing interests

M.C.A. reports advisory board participation, honoraria and research funding to institution from Merck Sharp and Dohme, and contract research for BMS Australia, all outside the submitted work. M.C.A. and J.A.W. are co-inventors on patent applications "Methods and compositions for treating cancer" (WO2020106983A1) and "Methods for improving sex-dimorphic responses to targeted therapy in melanoma" (US20200164034A1), unrelated to the current work. M.A.D. has been a consultant to Roche/Genentech, Array, Pfizer, Novartis, BMS, GSK, Sanofi-Aventis, Vaccinex, Apexigen, and ABM Therapeutics, and has been the PI of research grants to MD Anderson by Roche/Genentech, GSK, Sanofi-Aventis, Merck, Myriad and Oncothyreon. J.A.W. is an inventor on patent applications submitted by the University of Texas MD Anderson Cancer Center "Methods for enhancing immune checkpoint blockade therapy by modulating the microbiome" (PCT/US17/53.717) and "Targeting B cells to enhance response to immune checkpoint blockade" (UTSC.P1412US.P1-MDA19-023); reports compensation for speaker's bureau and honoraria from Imedex, Dava Oncology, Omniprex, Illumina, Gilead, PeerView, Physician Education Resource, MedImmune and Bristol-Myers Squibb; serves as a consultant/advisory board member for Roche/Genentech, Novartis, AstraZeneca, GlaxoSmithKline, Bristol-Myers Squibb, Merck, and Ella Therapeutics. AJL reports personal fees from Merck, Bristol-Myers Squibb, Novartis, and Roche/Genentech; personal fees and non-financial support from ArcherDX and Beta-Cat; grants and non-financial support from Medimmune/AstraZeneca and Sanofi; and grants, personal fees and non-financial support from Janssen, all outside the submitted work. J.E.G. reports consulting and/or advisory roles to Merck, Novartis, Syndax, Bristol-Myers Squibb, Regeneron, unrelated to this work. P.H. reports consulting/advisory roles for Dragonfly, Immatics, GSK, and Sanofi. C.B. reports consulting for and receiving funding from Iovance Biotherapeutics and Obsidian Therapeutics as well as serving as a consultant/advisory board member for Myst Therapeutics and Turnstone Biologics. All other authors report no relevant disclosures.
