## [Peer Review File · Nature Communications]

Reviewers' Comments:

Reviewer #1:

Remarks to the Author:

This manuscript by Andrews et al., presents a comprehensive molecular characterization of melanoma cell intrinsic states by multi-omics analysis of a panel of 68 early passage cell lines derived from human melanoma metastases. Based on whole transcriptomic analysis, three clusters of cells were identified representing what the authors called melanoma cell transcriptomic states (MCS), which differentially expressed melanocytic (MEL) and a mesenchymal-like (MES) gene sets. These two gene signatures were able to distinguish melanomas from TCGA data set with specific immune gene expression profiles. An extensive analysis of mRNA, microRNA and long non-coding RNA (lncRNA) expression, somatic mutations, copy number alterations (CNA), and DNA methylation profiles revealed highly diverse regulatory mechanisms of each MCS in the cell lines (their 68 panel and CCLE melanoma panel) as well as the melanomas from TCGA.

Overall, this is a descriptive work that presents a large amount of data related to the regulation of melanoma differentiation states in human cell lines and tumors, providing valuable resources to the melanoma research community. However, no functional validation of the findings from the multi-omics analysis is shown and I also have concerns about how these results could translate into a better prediction of patient outcome after treatment. The main questions I think should be addressed are below:

1) How much overlap exists between the melanoma cell transcriptomic states (MCS) identified by the authors and other melanoma differentiation signatures previously reported? MEL and MES gene signatures should be compared, for example, with Mitf-high vs Axl-high signatures (Tirosh et al., 2016, DOI: 10.1126/science.aad0501) or proliferative vs invasive signatures (Hoek et al., 2008, DOI: 10.1158/0008-5472.CAN-07-2491; or Verfaillie et al., 2015, DOI: 10.1038/ncomms7683). Would the 68 cell lines described in this study be clustered/classified in the same manner using those other signatures?

2) In general, cluster 2 (INT state) needs a more detailed characterization at transcriptional level. It is not justified in the manuscript why the INT state is defined by the intermediate expression of MEL and MES genes instead of using the differential genes expressed by cluster 2. That limit the identification of this state, especially in tumor samples, as was observed for TCGA data set. A comparison of the DE genes in this cluster with the additional melanoma differentiation stages recently reported (Tsoi et al., 2018, DOI: 10.1016/j.ccell.2018.03.017; and Rambow et al., 2018, DOI: 10.1016/j.cell.2018.06.025) would provide insightful information about the phenotype of the cells (and tumors) from this cluster.

3) Several miRNAs and lncRNAs were identified that potentially could regulate key genes of the MCS, such TRPM1, EDNRB, ZEB1, etc. The authors should functionally validate the direct regulation of these genes by those miRNAs and lncRNAs in representative cell lines of each state. Similarly, no validation of the direct impact of the DNA methylation results on gene expression is presented.

4) The authors show that few genes from MEL or MES sets overlap with MCP-counter endothelium and fibroblast gene signatures (Suppl fig 3c). Is there any overlap with other immune cell signatures? For example, it is well known that Axl is expressed by myeloid cells, including M2-like macrophages, which could influence the classification of melanomas as MES-like. Is it possible to estimate the expression of MEL and MES genes specifically by the melanoma cells from the bulk RNAseq of the tumors or correct the potential expression from stromal/immune cells as done by Tsoi et al., (DOI: 10.1016/j.ccell.2018.03.017)?

Additional evidences of the correlation of the MCS and the immune microenvironment are needed, for example, by analyzing the expression of the T cell exclusion signature (specifically expressed by melanoma cells) recently published by Jerby-Arnon et al. (DOI: 10.1016/j.cell.2018.09.006) in the 68 and CCLE cell lines.

5) The potential of MEL and MES gene signatures to predict patient outcome after treatment is not clear from the analysis presented. Is the expression of the signatures correlated with patient overall survival in the cohort from which the 68 cell lines were generated or in the TCGA data set? The low number of patients receiving the same therapy in this cohort doesn't allow to determine whether the trend observed of worst clinical benefit for MES state is general or more significant for specific

treatments. Recently, a melanoma plasticity signature has been significantly correlated with the response to immune checkpoint blockade antibodies (Perez-Guijarro et al., 2020, DOI: 10.1038/s41591-020-0818-3). However, MEL and MES states showed poor association with the response to Anti-PD-1 in Riaz data set. That could be explained because both pre- and on/post-treatment samples were analyzed together while significant increase of MES state was observed in paired samples after treatment. To better understand MCS signatures predictive value, their correlation with therapeutic response and patient survival should be analyzed in additional data sets, and pre- and post-treatment samples should be also analyzed separately. In addition, representative cell lines of each state could be treated with targeted or chemotherapies to obtain more direct evidence of the contribution of MCS to drug resistance.

Minor points:

- 1) A more detailed description of the clinical characteristics of the patient cohort presented, such as treatment received, response, time until progression and/or survival after treatment is needed.
- 2) The authors claim that the CNA profiles of 68 and CCLE cell lines "indicated the existence of a distinct INT state rather than these samples representing a transitional or mixed state." However, the CNA profiles of INT cells in both panels do not look the same. A detailed summary of the unique CNA found in INT state cells would help to interpret these data.
- 3) In the discussion is stated that "The relatively lower number of MES samples in our own patient cohort, as well as from the TCGA melanoma cohort, may thus reflect a lower frequency of cancer cells that are able to escape the MEL set-point to become the dominant population in melanoma tumors." However, all samples analyzed were from metastatic lesions. Is there any similar distribution of melanoma phenotypes reported for primary melanomas? The authors should mention that in the discussion.
- 4) In figure 2a, the tissue of origin of the samples analyzed should be indicated. Any significant association of the MCS and/or MCP-counter cell populations with the tissue of origin should be mentioned.
- 5) Fig. 7d: for a better visualization the MEL and MES dots of INT cluster should align with the corresponding box.

Reviewer #2:

Remarks to the Author:

Multi-modal molecular programs regulate melanoma cell state and clinical outcomes

In this manuscript, Andrews et al. aimed at mapping multi-modal regulatory programs governing specific transcriptomic state, which may be linked to biological and clinical relevance. Using "ex vivo" and "early-passage" melanoma cell lines from metastatic melanoma tumors, authors defined 3 melanoma cells transcriptomic states (MCS), MEL, INT and MES and claimed that these states are regulated by multiple genetic and epigenetic programs which were found to be conserved in patient samples (based on TCGA information). Authors further claim that these transcriptomic states are associated with immune features and correlate with checkpoint inhibition. While the manuscript consists of large amount of descriptive information, conclusions often made appear to be speculative, lacking supportive data. Overall conclusions, as much as attempt to establish a cohesive picture, need major overhaul and substantial support by data.

Major concerns

1. Whole study is based on the multiplatform analysis using "early-passage melanoma cell lines" from metastatic patient tumors. Authors attempted to map 3 transcriptomic signatures in TCGA datasets (tumors) which were defined as highly conserved in patients' tissue samples (line 397 of page 17). However, no data is presented to support the claim that these signatures faithfully reflect those seen in parental tumors. For one, it is not clear what were the statistical / scientific criteria to establish similarity between transcriptomic MCS in cell lines and TCGA (tissues). Likewise, the statement made

in line 198 of page 47 “potentially reflecting an interaction between BRAF status and MCS regarding adaptability to growth in vitro, even at low passage” is not clear. Does this reflect possible adaption of tumor to culture conditions?

2. Authors claim for three phenotypic states with distinct MCS. However, no experimental data support how these states differ (proliferative vs. invasive, for example) and whether the claimed phenotypic difference is conserved in the corresponding parental tumors.

3. The MCS pattern in CCLE dataset resembles “early-passage cell line” dataset rather than that found in the TCGA dataset. Authors need to clarify the advantage of “early-passage melanoma cell lines” used in this study, instead of the routinely used melanoma cell lines. Further authors need to explain what is the basis for similarities found between the three MCS conventional melanoma cell lines and the “early-passage cell lines” (Fig. S5a).

4. The subset of MEL mentioned in line 156 of page 7 needs to be better detailed (Fig. 2d). What is the justification to focus on a subset of MEL-like samples, as claimed by authors?

5. Given the heterogeneity seen in Fig. 2a, authors should perform cross-validation using additional datasets, including the parental tumors used in this study. Here again, transcriptomic data from the parental tumors would be essential to establish authors claims.

6. Authors ought to provide more details about specific chromosomal segments that support a distinct INT state in both Fig. 3e and Fig. 5b. The CNA patterns of INT state in chromosomes 7, 8 and 10 of melanoma cell lines from CCLE (Fig. S5b) does not appear to resemble those seen in other states.

7. The claim for “overall dominance of CNA occurring in the MEL state” should be substantiated by data, using different datasets like CCLE used in this study.

8. In some genes (e.g., NUA1, HRH1), the pearson’s correlation of miRNA and gene expressions shows an opposite trend [e.g. negative in cell line (Fig. 5b), positive in TCGA (Fig. 5c)]. This needs to be clarified.

9. Authors reliance on distinct regulatory points (miRNA, lncRNA and methylation, etc) require to be substantiated by experimental-based validation.

10. With the bulk of RNAseq datasets, it is hard to exclude the role and possible involvement of stromal components. If transcriptomic profile of stromal with responsive phenotype is closer to MES, then, regardless of MCS on pre-treatment, responder transcriptome would be like MES. Authors needs to provide data and/or discuss to clarify / exclude this possibility. One wonders whether scRNAseq data would show more MES-type MCS in the responders.

Minor concerns

1. The statistically significant values of each cell should be displayed in the heatmap (Fig. 2a).

2. A better description required for Fig. 3c, line 168 of page 8.

3. Graphical mapping (illustration) of genes described in line 206 ~207 of page 9 would be helpful.

4. What are the biological implications for limited positive correlation between epigenetic age and chronological age seen in INT? Statistical p value needs to be presented with slope or rho value.

Reviewer #3:

Remarks to the Author:

Andrews et al. generated melanoma cell lines from patient-derived metastatic melanoma tumors and these cells were used for their multi-modal analyses with the combination with publicly available dataset. Firstly, authors classified melanoma cells into $K = 3$ based on the results of initial analyses. The melanoma cell transcriptomic states (MCS) are categorized as MEL (melanocytic), INT (intermediate) and those MCS showed specific gene expressions such as MITF, SOX10, and EDNRB for MEL, ZEB1, AXL, and EGFR for MES. Additionally, MCS are further investigated by using multiple omics data such as DNA methylation, CNA, miRNA, and lncRNA.

Here are my comments.

Major comments

1. RNA expression levels were normalized by RPKM and which is known to be inconsistent among samples (Bioinformatics. 2010, 26(4):493-500, Theory Biosci. 2012, 131(4):281-285 etc). It is highly suggested to use TPM for expression analysis. Actually, lncRNA was normalized to have TPM values in your study.
2. In Figure 2b and c, authors conducted t-test after ANOVA, which is incorrect. Please do appropriate post-hoc test whether the difference is statistically significant.
3. What is the precise definition of early-passage melanoma cell lines? Please describe more detailed in method section.
4. Some data (both main figures and supplementary figures) such as mutational signature analysis shown in Figure 3d bottom for example are not explained in the manuscript. Please explain or omit if unnecessary.
5. P14, line 322: The authors stated, "However, yet further layers of functional control are indicated, as evidenced by low correlation between EGFR transcript levels and EGFR phosphorylation at functionally critical tyrosine residues determined by RPPA (pY1068 Pearson rho 0.44, $p < 0.001$; pY1173 Pearson rho -0.019, $p = 0.55$; Fig. S8e-f)". It will be difficult to draw a conclusion based on the data in Fig. S8e-f because many studies demonstrated that there is an expression gap between RNA expression and protein expression. Therefore, correlation analyses between RNA expression and phospho-proteins are misleading.
6. P9, line 194, Author stated, "a higher frequency of BRAF hotspot events in the MEL samples ($p = 0.016$, Fisher's exact test) and of NRAS hotspot events in the MES samples ($p = 0.03$, Fisher's exact test), consistent with the typically mutually exclusive alteration of these two genes." I am curious to know the expression levels of those two genes. In other words, genes shown in Fig 3a-c are not frequently mutated judging from Fig 3d. How would you interpret these gaps between expression levels and gene mutations? One explanation could be as author stated in line 204, "CNA differences were observed between MCS, e.g. gain of chromosomes 1p, 7q, 15 (p+q), and losses of chromosomes 9p, 11 (p+q), 18p, 19q (Fig 3e)." However, MITF locates on Chr3, ZEB1 on Chr10, AXL on 19, SOX10 on Chr22, EDNRB on Chr13, and EGFR on Chr7p. None of the genes of interest mapped to the indicated gene locus, indicating that those gene expression levels might be regulated by other mechanisms, not CNA. This suggests that it should be indirect regulation, as TRPM1 and miR-211 locate on Chr15. In addition, authors stated in line 197, "In high-purity TCGA melanoma samples, BRAF mutations were not associated with MCS, potentially reflecting an interaction between BRAF status and MCS regarding adaptability to grow in vitro, even at low passages (Fig. S4)." If you conclude abovementioned, what is the advantage of using TCGA data in your analyses?
7. Does the MEL-MES state reflect the transition from MEL to MES, from MES to MEL, or truly mixture of both? P6, line 125, authors stated, "Cluster 2 was described as intermediate (INT), consistent with a transitional state in the phenotype-switching model of melanoma previously by others." If this is the case, does the MEL-MES state reflect the transition from MEL to MES, from MES to MEL, or truly mixture of both? However, author also stated in P10, line 216, "As in our own line, CNA profiles indicated the existence of a distinct INT state rather than these samples representing a transitional or mixed state." On top of that, P11, line 244, "Despite high data dispersion, linear regression revealed highly similar trends for both MEL and MES groups for which DNAmAge was nearly constant regardless of chronological age (slope = -0.0023 MEL, -0.023 MES). By contrast, epigenetic age of INT samples showed positive correlation with chronological age (slope = 0.50). Together with distinct CNA profiles, these findings add further support to the existence of fundamental molecular differences between the INT and either MEL or MES states." Initially author described the INT state as "transitional state". Later, described as it is not. Please explain how you would interpret logically.
8. Supplementary Figure S9e-f, which groups showed a statistically significant? How did you analyze

to obtain p-values?

Minor comments

1. There is no color represented in Supplementary Figure S3A, color key box. Please fix it.
2. Format; please carefully review to amend the manuscript, e.g. p has to be *p* (italic format) and it should be shown with a following blank like $p = 0.01$ in Nature communications. There are two descriptions, one is 13 and the other is thirteen, for another example. Please amend all of them.
3. Why do the authors proceed with low RIN samples (2350, 2417, 2734) for miRNA sequencing? Is there any specific reason why authors did not redo extracting RNAs?
4. Molecular insight to the metastatic melanoma is weak. Authors performed RNA-expression analysis, DNA methylation analysis, GO analysis and so on but ended up in a possibility. Since authors generated cell lines, all or at least some of the important findings described in the manuscript should be confirmed by experimentally. For instance, P12, line 23, author stated, "While high transcript levels of ZEB1 in MES cells were associated with DNA methylation at gene body site, comparable of methylation in INT samples may be moderated by lower levels of LINC00702 in these cells, leading to increased availability of the ZEB1-targeting miRNA-4652-3p, thus reducing ZEB1 expression 56 (Fig. S7d, Fig. 4c)." It is good to validate the miRNA expression levels by northern-blotting and protein expression levels by western blotting, or other reasonable methodologies.
5. As this paper mentioned (BMC cancer, 2015; 15: 242), there are no standardized log2 ratio cut-offs to define low-amplitude copy number gain/loss and high amplitude amplification/deletion. Based on the available published literature, author can consider to use a log2 ratio cut-off of ± 0.25 to define copy number gain/loss and a cut-off of ± 0.8 to define amplification and deletion. Please double-check the term; loss or deletion, gain or amplification. Related that, please amend the methods section "Somatic copy number alteration and purity analysis"
6. Please state in the manuscript when publicly available dataset was downloaded and used for analyses, e.g. TCGA-SKCM and others as author did state in CCLE dataset (accessed October 18, 2020).
7. Please give more details of both random forest and support vector machine's hyper parameter settings as well as obtained results. Also, please describe the result of SVM performance since none of the information is available in the main text but stated in the methods section.

REVIEWER COMMENTS

Reviewer #1 (Remarks to the Author):

This manuscript by Andrews et al., presents a comprehensive molecular characterization of melanoma cell intrinsic states by multi-omics analysis of a panel of 68 early passage cell lines derived from human melanoma metastases. Based on whole transcriptomic analysis, three clusters of cells were identified representing what the authors called melanoma cell transcriptomic states (MCS), which differentially expressed melanocytic (MEL) and a mesenchymal-like (MES) gene sets. These two gene signatures were able to distinguish melanomas from TCGA data set with specific immune gene expression profiles. An extensive analysis of mRNA, microRNA and long non-coding RNA (lncRNA) expression, somatic mutations, copy number alterations (CNA), and DNA methylation profiles revealed highly diverse regulatory mechanisms of each MCS in the cell lines (their 68 panel and CCLE melanoma panel) as well as the melanomas from TCGA.

Overall, this is a descriptive work that presents a large amount of data related to the regulation of melanoma differentiation states in human cell lines and tumors, providing valuable resources to the melanoma research community. However, no functional validation of the findings from the multi-omics analysis is shown and I also have concerns about how these results could translate into a better prediction of patient outcome after treatment. The main questions I think should be addressed are below:

1) How much overlap exists between the melanoma cell transcriptomic states (MCS) identified by the authors and other melanoma differentiation signatures previously reported? MEL and MES gene signatures should be compared, for example, with Mitf-high vs Axl-high signatures (Tirosh et al., 2016, DOI: 10.1126/science.aad0501) or proliferative vs invasive signatures (Hoek et al., 2008, DOI: 10.1158/0008-5472.CAN-07-2491; or Verfaillie et al., 2015, DOI: 10.1038/ncomms7683). Would the 68 cell lines described in this study be clustered/classified in the same manner using those other signatures?

We thank the reviewer for the excellent suggestion of including more explicit comparisons with other key 'signatures' that have been reported in the literature. We have evaluated the overlap between the gene sets described (MDACC, Hoek, Tirosh and Verfaillie), noting that there are important differences in methodology, as well as the objectives motivating each study, which for context are summarized as follows:

- Hoek and colleagues performed microarray analyses on multiple distinct melanoma cell line sets. Hierarchical clustering of differentially expressed genes separated the melanoma cell line samples in each set into three major groups, with two predominant transcriptomic states representing the ends of a spectrum with distinctly different metastatic potentials. Genes involved in melanocyte differentiation were upregulated in the weakly metastatic melanoma cohort, but downregulated in the strongly metastatic cohort, while genes involved in modifying extracellular environments were highly expressed in the latter. These two transcriptomic phenotypes were labeled 'proliferative' and 'invasive', respectively, and were validated in melanoma tumors. Genes that were highly abundant in the proliferative phenotype were associated with MITF and SOX10, while genes that were highly abundant in the invasive phenotype were associated with TGF β signaling
- Tirosh and colleagues performed single cell RNA sequencing and exploratory analysis using unbiased dimensional reduction (tSNE) and principal component analysis (PCA) within cells classified as malignant on the basis of inferred aneuploidy. The first principal component (PC1, i.e. the dominant source of variation between samples) was attributed to technical factors. PC2-6 were considered to represent meaningful biological variation. MITF was highly correlated with PC4 and 5 and thus intrinsically not responsible for most of the variation observed in the malignant cells. Nonetheless, two gene sets were derived across all cells (i.e. irrespective of unbiased cluster memberships considering the entire transcriptome); one

defined as positively correlated with MITF, and one defined as inversely correlated with MITF. AXL was one of many genes that were inversely correlated with MITF although this does not imply that AXL is an obligate driver of the MITF-low condition

- Verfaillie and colleagues performed differential gene expression using standard statistical methods and thresholds (DESeq2, $|\log_2FC| \geq 1$, $p_{adj} < 0.05$) between invasive (n=2) and proliferative (n=9) cell lines but it is not clear exactly how cell lines were classified (other than manual curation) and this strategy inherently excludes the possibility that cell lines were more appropriately assigned to some other state than a binary choice of invasive and proliferative. The gene lists are large and include prominent representation of non-coding RNA species, thus substantial non-overlap with any other published signature is evident

Overlaps between all of these signatures are shown in **Supplementary Figure 1g** (copied below) and accompanying text has been added to the manuscript (**line 175 in tracked changes version, line 163 in cleaned version**). Overlap regions with the MDACC cell line derived MEL/MES genes are color-coded for the relative proportion of MEL or MES representation within each overlap region as more red or blue, respectively. Overlaps with the Tirosh set are heavily driven by MEL genes; the relative lack of MES genes in the overlap suggests that many of the MITF anti-correlated genes in the Tirosh set are uninformative. Overlaps with the Hoek gene set are fairly balanced for MEL and MES representation, consistent with both strategies considering the extremes of the phenotypes for the purposes of differential expression. Overlaps are also fairly balanced with the Verfaillie gene sets, acknowledging that there are no overlaps for the vast majority of genes in the latter with any other signature due to the sheer size of the Verfaillie gene sets and content of non-coding RNA species.

Supplementary Figure 1g:

As suggested, we re-clustered our cell line panel based on expression of these other gene sets not including non-coding, deprecated, or pseudogene content or where no clear mapping between discrepant annotations existed. Our own and the Tirosh gene lists produced optimized 3 cluster solutions using consensus NMF, whilst Hoek and Verfaillie produced 4 cluster solutions, having 2 MEL-like and 2 MES-like subclusters, respectively. Overall consensus across these independent solutions was very good, as shown in **Supplementary Figure 2a** (copied below) and accompanying text (**line 184 in tracked changes version, line 172 in cleaned version**), consistent with a common underlying biology rather than gene signature redundancy given the relatively minimal overlap of specific genes included in these signatures.

Supplementary Figure S2a:

2) In general, cluster 2 (INT state) needs a more detailed characterization at transcriptional level. It is not justified in the manuscript why the INT state is defined by the intermediate expression of MEL and MES genes instead of using the differential genes expressed by cluster 2. That limit the identification of this state, especially in tumor samples, as was observed for TCGA data set. A comparison of the DE genes in this cluster with the additional melanoma differentiation stages recently reported (Tsoi et al., 2018, DOI: 10.1016/j.cell.2018.03.017; and Rambow et al., 2018, DOI: 10.1016/j.cell.2018.06.025) would provide insightful information about the phenotype of the cells (and tumors) from this cluster.

We agree that our initial characterization of the third (non-MEL, non-MES) state was lacking, leading to some ambiguity about its biological significance, and thank the reviewer for their suggestion of additional analyses to address this. In doing so, it is again important to review the rationale and strategy behind the cited datasets.

Tsoi et al., performed consensus clustering of 53 cell lines that had undergone RNAseq gene expression analysis and identified 4 clusters. The SubMap algorithm was used to compare their clusters with the original 3 Hoek cell line clusters, but precise methodology is unclear. Importantly, the Tsoi study was specifically interested in the nature of adaptive MAPK pathway blockade resistance and utilized a panel with 87% *BRAF* or *NRAS* mutated cell lines, which is higher than typical patient populations, Hoek's cell line sets, and our own panel. These clusters were projected onto a principal component space derived from an empirical *in vitro* differentiation program with some degree of objective fit. Differentially expressed genes between the clusters were identified to form the 7 signatures.

To address relationships between clusters in the Tsoi dataset and in our own, we emulated the cross-cluster mapping between the Tsoi transcriptomic dataset and our own cell line transcriptomic data, using the same SubMap algorithm Tsoi employed when comparing to Hoek's subtypes. These results are shown in **Supplementary Figure 1d** (copied below) and text (**line 151 in tracked changes version, line 139 in cleaned version**), and confirm that our "MEL" state is highly and specifically associated with the Tsoi "melanocytic" state. Our "MES" state is highly associated with the Tsoi "undifferentiated" state, and weakly with the "neural crest like" state. Our cluster 2, which we have now termed "neural-plastic", is highly associated with both the Tsoi "neural crest like" and "transitory" states. Thus overall our results are very consistent with those reported previously by Tsoi.

Supplementary Figure 1d:

Focusing specifically on cluster 2 as suggested by the reviewer, we evaluated the overlaps in the lists of genes differentially expressed between cluster 2 vs the other clusters in our own data, and the Tsoi gene signatures, and used class-label permutation tests to determine significance. These results are shown in **Supplementary Figure 1e** (copied below) and text (**line 158 in tracked changes version, line 146 in cleaned version**). Genes enriched in cluster 2 (formerly called “intermediate”) showed overlaps of at least 5% only with the “transitory”, “transitory-neural crest cells” and “neural crest cells” signatures of Tsoi. Genes downregulated in cluster 2 showed a minor but significant overlap only with the “melanocytic” signature. Overall, these findings further support the assignment of a transitory state with neural crest features to our cluster 2, which we now call a “neural-plastic” state.

Supplementary Figure 1e:

	overlap fraction cluster 2 UP genes	overlap fraction cluster 2 DOWN genes
melanocytic (M)	0.0% overlap $p = 1$	3.2% overlap $p = 0.013$
(T-M)	0.8% overlap $p = 1$	0.0% overlap $p = 1$
transitory (T)	10.0% overlap $p = 0.11$	0.0% overlap $p = 1$
(N-T)	52.0% overlap $p = 0.001$	0.0% overlap $p = 1$
neural crest like (N)	6.1% overlap $p = 0.29$	0.0% overlap $p = 1$
(U-N)	0.0% overlap $p = 1$	0.0% overlap $p = 1$
undifferentiated (U)	0.0% overlap $p = 1$	0.0% overlap $p = 1$

Rambow *et al.* primarily performed single cell RNA sequencing studies on a small number of patient-derived xenograft models to evaluate longitudinal transcription patterns in surviving melanoma cell phenotypes with the aim being to describe melanoma minimal residual disease. As best as we could determine from the details provided, highly variable genes across the single cell transcriptomic data were used to cluster the cells using NMF. The clusters thus defined (?n=10) were used for differential gene expression analysis (?within or across each time point) from which the top 100 candidates (?per cluster/time point or pooled) underwent pathway & transcription factor target analysis followed by manual curation and collapsing into 6 final gene signatures, effectively

producing curated metasignatures containing a total of 284 genes. However, 8 metasignatures are provided including 323 unique genes in the supplementary data. Due to the intervening step(s) of functional annotation and manual curation, it is not clear exactly how these signatures relate to the original unsupervised clusters identified, the treatment time points, or the four main described 'drug tolerant' states. Nevertheless, we utilized all 8 signatures provided to perform gene set enrichment analysis (GSEA) across the three clusters of our own cell line dataset using a (cluster vs rest) strategy and report these results in the text (**line 161 in tracked changes version, line 149 in cleaned version**):

- Cluster 1 "MEL": enriched for MITF, mitosis, pigmentation, immune signatures at $q < 0.001$ and hypometabolic signature at $q < 0.005$
- Cluster 2 "other": enriched for neuro signature at $q < 0.001$
- Cluster 3 "MES": enriched for invasion signature at $q < 0.001$. Trend for neuro signature at $q = 0.274$

We also evaluated the cluster 2 enriched ($n=78$) or depleted genes ($n=6$) versus the other clusters on their own merits for functional and structural enrichments:

- KEGG pathway enrichments (pooling all DE genes): sphingolipid metabolism, axon guidance, JAK-STAT signaling pathway, VEGF signaling pathway, BCR signaling pathway, focal adhesion, apoptosis, pancreatic cancer, small cell lung cancer (all $q = 0.0613$), TCR signaling pathway ($q = 0.0845$)
- chromosomal location using Positional Gene Enrichment analysis (<http://silico.biotoul.fr/pge/index.php>): no striking chromosomal location enrichments were observed (e.g. large regions with small enrichment or small regions of high enrichment)

3) Several miRNAs and lncRNAs were identified that potentially could regulate key genes of the MCS, such TRPM1, EDNRB, ZEB1, etc. The authors should functionally validate the direct regulation of these genes by those miRNAs and lncRNAs in representative cell lines of each state. Similarly, no validation of the direct impact of the DNA methylation results on gene expression is presented.

Among the MCS genes, ZEB1 has substantial support as a driver of invasive malignant melanoma (Caramel *et al.*, Cancer Cell 2013, doi: 10.1016/j.ccr.2013.08.018). We confirmed that ZEB1 abundance is anti-correlated with miR-211 at both the transcript and protein levels in both cell lines and melanoma tissues:

Reviewer Figure R1Q3a:

We performed RT-qPCR analysis to validate the differential miR-211 versus ZEB1 expression that was observed by RNA-Seq:

Reviewer Figure R1Q3b:

We performed exogenous expression experiments in which miR-211 was expressed in melanoma cell lines with endogenously high ZEB1. The introduction of miR-211 markedly reduced ZEB1 transcript abundance and corresponding protein levels compared to control transfection:

Reviewer Figure R1Q3c,d:

These *in vitro* experiments validate the anti-correlation of miR-211 and ZEB1 we observed in melanoma cells and support the result we generated, using three computationally derived miR target databases and two experimental miR databases, that miR-211 targets ZEB1. Please also see the response to question 5) below concerning the association of DNA methylation with MCS cell profiles.

4) The authors show that few genes from MEL or MES sets overlap with MCP-counter endothelium and fibroblast gene signatures (Suppl fig 3c). Is there any overlap with other immune cell signatures? For example, it is well known that *Axl* is expressed by myeloid cells, including M2-like macrophages, which could influence the classification of melanomas as MES-like.

The reviewer speaks to the known issue of confounding/contamination inherent to all transcriptome-based immune deconvolution procedures. We reported that overlap between MEL and MES gene signatures was only observed with the MCP-counter endothelium and fibroblast signatures which remains the case with our updated MEL/MES gene lists derived from TPM data analysis. We initially intentionally did not apply immune deconvolution algorithms to our cell line data, based on the principle that melanoma cell lines (should) contain little to no immune cells and should therefore register essentially no immune cell content in any of these algorithms; in other words, any immune/stromal cell deconvolution in cell lines is essentially spurious. However, to address the reviewer's point about potential overlap with other published immune signatures, we applied four highly cited algorithms to our melanoma cell lines using the `immunedconv` package in R to determine which cell subset signals appear to be potentially spurious in melanoma samples. Specifically, we used EPIC and `quantIseq` (both utilize a deconvolution matrix methodology and produce absolute cell fractions), MCP-counter (marker gene expression methodology, inter-sample relative quantification only, not previously performed on our cell lines), and TIMER (deconvolution

matrix methodology, inter-sample relative quantification only) in order to survey a variety of methods and outputs for which simple gene list overlaps do not fully reflect the accuracy of inferred cellular content using such tools. These results are shown in **Supplementary Figure 4** (copied below) and text (**line 251 in tracked changes version, line 230 in cleaned version**) and confirm that stromal and myeloid signatures are highly inaccurate in melanoma samples, with estimation of cancer associated fibroblast scores being particularly unreliable in MES cluster melanoma samples.

Supplementary Figure 4:

Is it possible to estimate the expression of MEL and MES genes specifically by the melanoma cells from the bulk RNAseq of the tumors or correct the potential expression from stromal/immune cells as done by Tsoi et al., (DOI: 10.1016/j.ccell.2018.03.017)?

We thank the reviewer for this suggestion. Tsoi attempted to minimize contamination of the melanoma cell transcriptomic signature by excluding genes from consideration that were presumed to have immune cell origin based on high correlation with ordinated gene expression data of “immune” genes taken from innateDB and “keratin” genes being genes annotated as keratins in the NCBI gene database. It is important to note several caveats with this strategy:

- this is distinct from attempting to apportion some of the gene expression signal to a melanoma cell or an immune cell or a connective tissue cell origin
- high correlation with an “immune gene signature” does not necessarily imply that a gene is expressed principally in an “immune” cell nor does it suggest that such genes lack highly relevant functions in melanoma cells that contribute to melanoma cell phenotype and/or behavior, thus this “exclude-all-immune-gene-correlated-genes” strategy is inherently at risk of removing genuinely relevant genes from consideration, an issue which was acknowledged and specifically addressed in Jerby-Arnon *et al.*

- the innateDB gene set is manually curated and constantly evolving thus it is not possible to be certain that we have used the same input list of ‘immune’ genes. Additionally, innateDB is concerned primarily with innate not adaptive immunity, yet the bulk of immune cell infiltrate that could contaminate transcriptomic signatures in melanoma tumor samples may well be adaptive, not innate
- it is stated that ROC curve analysis was used to optimize the correlation value threshold above which a gene correlated (presumably positively) with the PCA-derived immune signature score was removed from consideration. However, it is not apparent how such a threshold optimization could be performed in the absence of an ‘absolute truth’ regarding whether an immune score-correlated gene really is, or isn’t, an immune gene that acts only as a confounder in transcriptomic data, thus we were unable to perform this ‘optimization’ step in our analogous procedure

We took the unique set of genes currently curated in the innateDB gene set and used expression of these genes in TCGA SKCM samples (n=471) to perform principal component analysis after filtering for very low-expressed genes (TMM count <1 in more than 30% of the samples). The first principal component was taken as the “immune signature score” and used to identify genes in the remainder of the transcriptome (considering only protein-coding or transcribed/translated pseudogenes):

- Of the 305 genes positively correlated with the immune signature score (r value >0) and statistically significant (p value <0.05), the strongest correlation was relatively weak (r value 0.33) and the top-most correlated genes included many mitochondrial genes which may indicate a contaminating effect of necrotic cellular nucleic acid. Only 9 of the immune signature correlated genes overlapped with marker genes in our MEL/MES gene sets: *BIRC7* (r=0.15), *GPR143* (r=0.11), *OCA2* (r=0.22), *CA14* (r=0.16), *TRPM1* (r=0.12), *HES6* (r=0.09), *MFSD12* (r=0.13), *TSPAN10* (r=0.18), *PMEL* (r=0.17). It seems likely that few to none of these would be removed as confounders if an r value threshold were applied, notwithstanding the unclear approach taken to define an r value threshold beyond an arbitrary value (e.g. r>0.6 for an at least moderately strong correlation)
- Taking the most conservative approach and excluding all of these genes from consideration, we performed cNMF based clustering of all 471 TCGA SKCM samples using expression of either the full MEL/MES marker gene list (“fullsig.clusters”, optimal rank=3) or the ‘short’ list excluding the above genes (“shortsig.clusters”). The results (see **Reviewer Figure R1Q4.1**, below) show relatively minimal impact on the cluster assignments
- Overall, the full marker gene list assigned 157 MEL, 199 neural/plastic (NPLAS) and 115 MES samples, with 13%, 39% and 6% “mismatched” when using the short marker gene list, respectively. Mismatches show that the dominant effect of removing immune signature correlated genes was to reclassify 69 NPLAS samples as MEL, and 14 MEL samples as NPLAS. This loss of MEL-NPLAS definition is an expected result given the exclusion of the genes *OCA2* and *PMEL* (well-known to characterize the melanocytic differentiation state) and *TRPM1* which we identified to be particularly relevant to the classification of the NPLAS state in our cell line analysis

Reviewer Figure R1Q4.1:

Additional evidences of the correlation of the MCS and the immune microenvironment are needed, for example, by analyzing the expression of the T cell exclusion signature (specifically expressed by melanoma cells) recently published by Jerby-Arnon et al. (DOI: 10.1016/j.cell.2018.09.006) in the 68 and CCLE cell lines.

Jerby-Arnon et al. developed a generically applicable strategy for identification of biologically interesting signatures in bulk transcriptome data leveraging findings from single cell datasets and applied this to melanoma samples. Importantly and by design, the final T cell exclusion signature includes genes that are not specifically expressed only in melanoma cells. We applied the full UP and DOWN exclusion signature (not the seed signatures) to the CCLE melanoma cell lines and calculated the T cell exclusion Overall Expression (OE) scores using the code supplied by Jerby-Arnon *et al.* We also implemented an independent but conceptually similar approach to derive a T cell exclusion signature score by maximizing the signed sum of UP and DOWN signature gene expression values from mean-centered and scaled data, arriving at very similar results. These are shown at gene expression level for the UP and DOWN signature genes:

Reviewer Figure R1Q4.2a:

TCE scores by cluster are summarized below (left, original method; right, analogous method).

Reviewer Figure R1Q4.2b:

The results indicate highly variable scores amongst samples of each class, however as a group the MES cell lines have the lowest T cell exclusion score values but there are no statistically significant between-group differences.

Although the actual score values are not directly meaningful, they provide an indication of relative expression of the T cell exclusion module which is designed to correlate with the transcriptome-inferred T cell content of a bulk tumor. However, it must be noted that the signature genes were constructed from bulk and dissociated tumor material such that the full signature should correlate

positively with the seed signature of the same directionality, and should also anti-correlate with the seed signature of opposite directionality. As seen in plots of these relationships below, these relationships do not entirely hold true, suggesting that this TCE signature does not perform well or is of low applicability to these high-passage CCLE cell lines.

Reviewer Figure R1Q4.2c:

In our own panel of melanoma cell lines we found better performance of the T cell exclusion signature as evidenced by expected direct correlations between the full and seed signature lists in both directions, and expected inverse correlation between the full DOWN signature and the seed UP signature as shown below. However, the full UP signature demonstrated no relationship to the seed DOWN signature, contrary to design, suggesting that although this tumor-derived signature may be more applicable to early passage cell lines than late passage cell lines, its performance in cell lines may not be robust in general and may be reflective of a degree of transcriptomic drift *in vitro* due to the absence of T cell infiltration in culture.

Reviewer Figure R1Q4.2d:

T cell signature scores per cell line were variable and overlapping across the cell line panel, with no statistically significant relationship to MEL/MES cluster as shown below.

Reviewer Figure R1Q4.2e:

5) The potential of MEL and MES gene signatures to predict patient outcome after treatment is not clear from the analysis presented. Is the expression of the signatures correlated with patient overall survival in the cohort from which the 68 cell lines were generated or in the TCGA data set?

We apologize for any confusion regarding these analyses. We evaluated the association between the transcriptomic cell state in cell lines and the clinical outcomes of the patients from whom those cell lines were derived, as we have complete knowledge of these patient's treatment histories and

outcomes. These results are presented in the results section “Clinical outcomes for melanoma sample donors are influenced by MCS” beginning **line 488 (tracked changes document; line 407 in cleaned version)** and associated results in Figure 7a-b, Supplementary Table 1, and Supplementary Figure 9a. As discussed in the text, formal stratified analyses of outcome considering each treatment modality separately were not possible due to the relatively low number of samples per individual treatment type. However, regardless of treatment type, MES samples had numerically inferior duration of clinical benefit than patients assigned to other MCS groups.

Survival analyses using TCGA SKCM (melanoma) samples are problematic as treatment data are not widely available, the survival durations indicated are not uniformly measured with respect to the date that the tumor (which was studied) was sampled, and/or the staging information provided applies to the time of first melanoma diagnosis, not the stage at the time the studied tumor was sampled. Thus, although a common practice, the assessment of tumor-derived biomarkers (e.g. gene expression/signatures) with survival outcomes in the full SKCM dataset is not truly possible and at best is confounded by other relevant clinical variables for which correction cannot be made.

The low number of patients receiving the same therapy in this cohort doesn't allow to determine whether the trend observed of worst clinical benefit for MES state is general or more significant for specific treatments.

This is exactly correct. We nonetheless feel that there is a compelling signature of therapeutically relevant impact, which is supported by existing literature in cytotoxic and MAPK targeted therapy at least, that motivates future, larger, studies to address this specifically, in the context of the modern standard-of-care treatment paradigm. We feel that similar concepts of cancer cell phenotype influencing therapeutic outcome are emerging as therapeutically relevant in immunotherapy, as suggested for example by recent data presented in abstract form showing that a stromal/TGF β /EMT signature predicted poor response to pembrolizumab in platinum-ineligible unresectable urothelial cancer patients, even after accounting for the T cell inflamed gene expression profile (Grivas *et al.* doi: 10.1200/JCO.2019.37.7_suppl.433 Journal of Clinical Oncology 37, no. 7_suppl (March 01, 2019) 433-433).

Recently, a melanoma plasticity signature has been significantly correlated with the response to immune checkpoint blockade antibodies (Perez-Guijarro et al., 2020, DOI: 10.1038/s41591-020-0818-3). However, MEL and MES states showed poor association with the response to Anti-PD-1 in Riaz data set. That could be explained because both pre- and on/post-treatment samples were analyzed together while significant increase of MES state was observed in paired samples after treatment. To better understand MCS signatures predictive value, their correlation with therapeutic response and patient survival should be analyzed in additional data sets, and pre- and post-treatment samples should be also analyzed separately.

We agree that a more complete understanding of the prognostic or – more importantly – predictive value of classifying melanoma cell molecular features is critical. Our primary motivation in this study was to explore the additional layers of molecular regulation that contribute to phenotype, beyond evaluation of the transcriptome. While we do present a valuable resource in the form of a unique patient cohort from whom these lines were generated, as noted the patient cohort is heterogeneous, and was never designed to answer treatment type-specific questions definitively, beyond the hypothesis-generating results presented.

We therefore utilized the Riaz dataset because it is anti-PD-1 specific and longitudinal. We analyzed samples separately at each time point, not just pooled, but apologize for any confusion given that not all permutations of the analyses were presented. There is a trend towards a difference in response category (binarized) across the three MCS clusters considering all samples ($p=0.081$, Fisher exact test), which is indeed driven by a higher proportion of responders in non-MEL-like samples at

the on-treatment time point ($p=0.0055$), as there is no significant difference considering only pre-treatment samples ($p=0.61$). Considering the earlier discussion regarding the confounded nature of stromal signatures in melanoma samples classified as MES-like, it is possible that on-treatment samples are subject to a degree of confounding wherein responding tumors are likely to have lower viable melanoma cell content and thus may be at higher risk of 'incorrect' classification as non-MEL-like. For this reason, and due to the more immediate clinical utility of a pre-treatment predictive biomarker, the lack of clear association with response in pre-treatment samples is considered more clinically relevant.

It is also worth noting that the Riaz dataset contains a mixture of melanoma histologies including cutaneous, acral, mucosal and uveal melanoma, and 7 patients are represented only at the on-treatment time point and are also not listed in the clinical annotation provided with the paper, thus it is possible that MCS classifications and responses have been further contaminated by these sources of heterogeneity. Indeed, when considering only cutaneous type melanomas for which full clinical annotation were provided, there was no significant difference in proportions of responder/non-responder patients at either pre-treatment or on-treatment time points, although there was a strong trend for the latter ($p=0.0505$).

Results (beginning **line 509 in tracked changes version, line 428 in cleaned version**) and figures (**Figure 7c,d; Supplementary Figure 9b-e**) have been updated to provide more detail about the Riaz cohort subsets.

In addition, representative cell lines of each state could be treated with targeted or chemotherapies to obtain more direct evidence of the contribution of MCS to drug resistance.

As suggested by the reviewer, representative cell lines of each state harboring a BRAF V600E mutation were treated with increasing doses of vemurafenib, a selective inhibitor of the mutated BRAF V600E kinase. Consistent with prior studies in which BRAF V600E cell lines with similar profiles to those we identify (e.g. Konieczkowski *et al.*, Cancer Discovery 2014;4(7):816-27; Müller *et al.*, Nature Communications 2014;5:5712), BRAF V600E MEL cells display marked sensitivity to BRAF inhibition (vemurafenib) at nanomolar concentrations, with near complete loss of viability at 1 μM as shown in **Supplementary Figure 1f**, copied below. BRAF V600E MES cells were resistant to vemurafenib at all concentrations tested, and similar resistance was observed in BRAF V600E mutant cells with a 'neural-plastic' gene expression profile.

Figure S1f:

In addition to targeted therapy, we tested the effect of the hypomethylating agent decitabine (5-aza-2'-deoxycytidine) in cell lines with the MEL versus MES gene expression profile, shown in **Supplementary Figure 7a** (copied below) and text (**line 345 in tracked changes version, line 307 in cleaned version**). MEL cells displayed a concentration-dependent decrease in cell viability, while

MES cells showed a modest decrease in cell viability across all concentrations, consistent with a more specific cytotoxic effect in MEL cells. Whether this is due to greater dependence on specific methylation events in MEL cells or a skewed balance of hypomethylation to hypermethylation events between the MCS remains to be determined.

Supplementary Figure 7a:

Minor points:

1) A more detailed description of the clinical characteristics of the patient cohort presented, such as treatment received, response, time until progression and/or survival after treatment is needed.

Additional clinical metadata has been added to the clinical metadata file (**Supplementary Table 1**) including lines of treatment immediately prior and post (if any) TIL tumor harvest, with relevant time intervals and response outcomes, as well as indication of prior exposure to various categories of systemic therapy.

2) The authors claim that the CNA profiles of 68 and CCLE cell lines “indicated the existence of a distinct INT state rather than these samples representing a transitional or mixed state.” However, the CNA profiles of INT cells in both panels do not look the same. A detailed summary of the unique CNA found in INT state cells would help to interpret these data.

The reviewer is correct that the differences in CNA profiles between states were not universal i.e. applicable across data/cell line sample sets. Our observation was primarily that examination of the CNA profiles between samples of the three clusters supported the independence of a distinct non-MEL/non-MES cluster (originally termed “intermediate” but in response to reviewer comments we have clarified this and now use the term “neural-plastic” or NPLAS state) and were not consistent with this NPLAS state simply being a mixture of cells in MEL or MES states. We did not mean to imply that the CNA profiles of the neural-plastic state were globally characteristic and would thus be identical in our own cell line panel and the CCLE panel, and by inference in any collection of melanoma cell lines. However, we acknowledge that the question regarding why the CNA profiles would be different is a very valid one. Copy number profiles have been described as sensitive to length of time *in vitro* (e.g. Ben-David *et al.*, Nature 2018;560(7718):325-330; Hughes *et al.*, Biotechniques 2007;43(5):575) although whether this is due to subclonal selection or global CNA evolution is not clear. Nonetheless, the significant ‘age’ difference between our cell line samples and those of the CCLE is likely a source of between-cohort CNA variation.

3) In the discussion is stated that “The relatively lower number of MES samples in our own patient cohort, as well as from the TCGA melanoma cohort, may thus reflect a lower frequency of cancer

cells that are able to escape the MEL set-point to become the dominant population in melanoma tumors.” However, all samples analyzed were from metastatic lesions. Is there any similar distribution of melanoma phenotypes reported for primary melanomas? The authors should mention that in the discussion.

This is an excellent point as one would expect that analogous variation in melanoma cell phenotype is seen across all stages of melanoma, although likely with differences in the relative proportions of more or less “differentiated” states. Indeed we do observe distinct clusters of samples based on MEL/MES gene expression in primaries just as we observe in metastatic tumors, noting also that the majority of metastases in the TCGA melanoma dataset are stage III nodal lesions thus the MEL/MES phenotypes are seen across all stages of melanoma (I/II/III/IV).

4) In figure 2a, the tissue of origin of the samples analyzed should be indicated. Any significant association of the MCS and/or MCP-counter cell populations with the tissue of origin should be mentioned.

These are the TCGA melanoma metastatic samples. Unfortunately, the provided metadata do not indicate source site per lesion beyond the annotation of “metastatic” which itself includes locoregional (in-transit or regional lymph node) or distant (nodal, soft tissue, or visceral) metastasis. We know from the original TCGA melanoma publication that the clear majority are from lymph nodes. To this point, the authors stated that:

“...we compared the expression of nine curated immune gene signatures (comprising 793 genes and detailed in Table S4B) in 172 samples from lymph nodes and 157 tumors from other tissues (Figures S5A and S5B). Reassuringly, there was no significant difference in expression of tested immune signatures between the samples from lymph nodes and non-lymph node tissues (Figure S5A), suggesting that the transcriptomic features of the immune subclass were not due to contaminating adjacent lymph node tissue” [Cancer Genome Atlas Network, Cell. 2015 Jun 18;161(7):1681-96]

thus we did not re-attempt to address this issue further.

5) Fig. 7d: for a better visualization the MEL and MES dots of INT cluster should align with the corresponding box.

This change has been made (corresponding to **Fig. 7b** in the current version).

Reviewer #2 (Remarks to the Author):

In this manuscript, Andrews et al. aimed at mapping multi-modal regulatory programs governing specific transcriptomic state, which may be linked to biological and clinical relevance. Using “ex vivo” and “early-passage” melanoma cell lines from metastatic melanoma tumors, authors defined 3 melanoma cells transcriptomic states (MCS), MEL, INT and MES and claimed that these states are regulated by multiple genetic and epigenetic programs which were found to be conserved in patient samples (based on TCGA information). Authors further claim that these transcriptomic states are associated with immune features and correlate with checkpoint inhibition. While the manuscript consists of large amount of descriptive information, conclusions often made appear to be speculative, lacking supportive data. Overall conclusions, as much as attempt to establish a cohesive picture, need major overhaul and substantial support by data.

Major concerns

1. Whole study is based on the multiplatform analysis using “early-passage melanoma cell lines” from metastatic patient tumors. Authors attempted to map 3 transcriptomic signatures in TCGA datasets (tumors) which were defined as highly conserved in patients’ tissue samples (line 397 of page 17). However, no data is presented to support the claim that these signatures faithfully reflect those seen in parental tumors. For one, it is not clear what were the statistical / scientific criteria to establish similarity between transcriptomic MCS in cell lines and TCGA (tissues). Likewise, the statement made in line 198 of page 47 “potentially reflecting an interaction between BRAF status and MCS regarding adaptability to growth in vitro, even at low passage” is not clear. Does this reflect possible adaption of tumor to culture conditions?

We appreciate the reviewer’s attention to the critical issue of concordance between patient tumor material and tumor-derived cell lines and implications this may have for any conclusions drawn in the opposite direction. This will be addressed more specifically in questions 2 and 5 below.

Regarding the appearance of similar clustering in the TCGA melanoma cohort, our statement is meant only to indicate that TCGA melanoma samples do cluster based on expression of the MEL/MES genes, or conversely that there is a gradient of expression between predominantly MEL through to predominantly MES, for which no statistical test is applicable.

The reviewer is correct to infer that we were theorizing that there may be a growth advantage under *in vitro* conditions for BRAF mutated melanoma cells, but we apologize if the wording was not entirely clear on this point. We note that there is an apparent excess of BRAF mutated cell lines in several published cohorts (ours, Hoek – Mannheim cohort of melanoma metastases 18/38=47% BRAF V600mt [Hoek *et al.* Pigment Cell Res. 2006 Aug;19(4):290-302], LM-MEL cohort 30/55=55% BRAF V600mt [Behren *et al.* Pigment Cell Melanoma Res. 2013 Jul;26(4):597-600]) compared with the TCGA melanoma (tumor) cohort (124/318=39% BRAF V600mt).

2. Authors claim for three phenotypic states with distinct MCS. However, no experimental data support how these states differ (proliferative vs. invasive, for example) and whether the claimed phenotypic difference is conserved in the corresponding parental tumors.

To experimentally test the phenotypic states in the MCS cells, proliferation and invasion assays were performed. Consistent with prior reports (Hoek, Ribas) MCS cells with a MEL gene expression profile exhibit far less invasion, but greater proliferation, compared to cells with a MES gene expression profile, whilst MES cell displayed the inverse as shown below. Notably, cells with an NPLAS gene expression profile showed varying levels of invasive and proliferative properties.

Reviewer Figure R2Q2:

The patient cohort from whom the cell lines were generated had quite variable treatment courses following their tissue harvest, making the assessment of tumor behavior relative to tissue profiling extremely difficult to elucidate. The tissues were harvested primarily for the clinical purpose of generating tumor infiltrating lymphocytes for potential future therapeutic use. At the time that the cell lines in our study were generated, having a sufficient amount of residual tissue for multi-omics analysis was rare. Thus, a comprehensive -omics comparison between the cell lines and parental tissues is also unfortunately impossible.

3. The MCS pattern in CCLE dataset resembles “early-passage cell line” dataset rather than that found in the TCGA dataset. Authors need to clarify the advantage of “early-passage melanoma cell lines” used in this study, instead of the routinely used melanoma cell lines. Further authors need to explain what is the basis for similarities found between the three MCS conventional melanoma cell lines and the “early-passage cell lines” (Fig. S5a).

We thank the reviewer for pointing out our over-emphasis of the early-passage status of the melanoma cell lines we generated and analyzed. Our intent was to simply assure readers of the relatively low passage status of the cell lines given the well-established genotypic and phenotypic changes that can occur due to selective pressures in longer-term culture environments and when cells become generations distant from the original cell culture isolate. It was not our aim in this study to attempt to differentiate advantages of early passage cell lines versus routinely used or commercially-sourced melanoma cell lines which in some cases may have been in culture cumulatively for many years if not decades. As the reviewer suggests, we similarly observe co-clustering of the CCLE melanoma cell lines relative to MEL and MES gene expression. These observations may speak to the highly intrinsic nature of these gene expression programs in melanoma cells, such that they are well preserved in independent cell line cohorts and tissues.

4. The subset of MEL mentioned in line 156 of page 7 needs to be better detailed (Fig. 2d). What is the justification to focus on a subset of MEL-like samples, as claimed by authors?

We apologize for the confusion which may have arisen from how this was worded. We were not referring to a subset of samples that we chose to focus on, rather we were attempting to point out that whilst MEL classified tumor samples have a lower median immune content than MES samples, T cells, CD8+ T cells and cytotoxic lymphocytes show a substantial range of expression, and are relatively highly expressed in about one-third of MEL samples. We believe highlighting this point is of value, as multiple studies report an association between CD8+ T cell enrichment in the tumor microenvironment and response to immune checkpoint blockade therapy, thus no single factor – including MCS – can be expected to be independently predictive in all cases. We have replaced “A subset of MEL samples...” with “Some MEL samples...” to make this less ambiguous (**line 234 in tracked changes version, line 217 in cleaned version**).

5. Given the heterogeneity seen in Fig. 2a, authors should perform cross-validation using additional datasets, including the parental tumors used in this study. Here again, transcriptomic data from the parental tumors would be essential to establish authors claims.

As outlined above, unfortunately a comprehensive comparison with ‘omics data from the parental tumors is not feasible due to the limited material that was available after these samples were utilized for their primary clinical purpose. We present cross-validation using the Riaz longitudinal anti-PD-1 treated cohort in **Figure 7c, d** and **Supplementary Figure 9b-e** which confirms similar patterns of immune infiltration across samples both prior to and after initiation of immune checkpoint blockade therapy; this dataset and analysis is also extensively discussed in the **response to Reviewer 1 Question 5** above. We have also extensively discussed the findings of the published T cell exclusion signature in response to **Reviewer 1 Question 4** above, which also shows heterogeneity.

6. Authors ought to provide more details about specific chromosomal segments that support a distinct INT state in both Fig. 3e and Fig. 5b. The CNA patterns of INT state in chromosomes 7, 8 and 10 of melanoma cell lines from CCLE (Fig. S5b) does not appear to resemble those seen in other states.

We appreciate the reviewer’s suggestion to add more detail to support these conclusions and agree that there are distinct segments of CNA as mentioned above. We have **updated Figure 3c** and the new **Supplementary Figure 6b** and associated text to this effect (**line 309 in tracked changes version, line 277 in cleaned version**).

7. The claim for “overall dominance of CNA occurring in the MEL state” should be substantiated by data, using different datasets like CCLE used in this study.

We are somewhat limited in our ability to use additional datasets to support a global conclusion of this type due to the need to have all modalities represented. However, what we intended to communicate with this statement was that copy number alterations, whether affecting MEL or MES genes, were more commonly due to alterations occurring in MEL cluster cells i.e. CN gains of MEL genes and/or CN losses of MES genes in MEL samples and have removed this potentially confusing wording (**line 305 in tracked changes version, line 273 in cleaned version**).

8. In some genes (e.g., NUA1, HRH1), the Pearson’s correlation of miRNA and gene expressions shows an opposite trend [e.g. negative in cell line (Fig. 5b), positive in TCGA (Fig. 5c)]. This needs to be clarified.

The reviewer is correct to note that not all miRNA-mRNA correlations were preserved (i.e. showed the same strength or directionality of correlation) in tumors as in cell lines. This was an entirely expected result because miRNA and mRNA are expressed in numerous cell types thus their abundance in bulk tumor data such as that of TCGA melanoma samples reflects the averaged

amount from all cell types present, not just that originating from melanoma cells. Accordingly, even without considering the impact of different *in vitro* versus *in vivo* microenvironmental signals (e.g. growth factors, extracellular matrix, etc) on gene expression, miRNA/mRNA signal “contamination” by non-melanoma cells present in tumor samples may weaken or even completely reverse the observed correlations for some miRNA-mRNA pairs. This phenomenon is expected for any molecular biomarker and underpins the principle of parallel study in both *in vitro* and *in vivo* systems.

9. Authors reliance on distinct regulatory points (miRNA, lncRNA and methylation, etc) require to be substantiated by experimental-based validation.

For experimental-based studies, please refer to our **response to Reviewer 2 Question 2** and corresponding figures, as well as the response to **Reviewer 1 Questions 3 and 5** and corresponding figures.

10. With the bulk of RNAseq datasets, it is hard to exclude the role and possible involvement of stromal components. If transcriptomic profile of stromal with responsive phenotype is closer to MES, then, regardless of MCS on pre-treatment, responder transcriptome would be like MES. Authors needs to provide data and/or discuss to clarify / exclude this possibility. One wonders whether scRNAseq data would show more MES-type MCS in the responders.

We agree with the reviewer that it is difficult when working with bulk tumor data to deconvolute the cellular source of transcriptomic (or even genomic, more generally) signals, and as a consequence it is also challenging to deconvolute the transcriptomic associations with clinically observable outcomes of the underlying biology. We have discussed this point extensively in results section “Stromal signatures confound interpretation of MCS in MES samples (**starting line 246 in tracked changes version, line 226 in cleaned version**). We also raise the same point as a major limitation to the ability of a transcriptome-based classifier to accurately classify bulk samples into MCS, particularly the MES group (**e.g. line 217 in tracked changes version, line 201 in cleaned version**). We also raise the possibility that this finding is an artefact of depletion of more “therapeutically susceptible” MEL cell content (**Discussion starting line 583, tracked changes version, line 498 in cleaned version**), as this could indeed lead to a MES-wise shift in transcriptomic features whether due to melanoma cell plasticity, selection, or stromal enrichment. Single cell or highly spatially-resolved techniques may indeed clarify this further in future studies of appropriate sample cohorts.

Minor concerns

1. The statistically significant values of each cell should be displayed in the heatmap (Fig. 2a).

These data have been added for each cell type.

2. A better description required for Fig. 3c, line 168 of page 8.

We have removed Supplementary Figure 3c (fibroblast/endothelium signature overlap) as it was of minimal additional value and adjusted wording in the paragraph indicated for clarity.

3. Graphical mapping (illustration) of genes described in line 206 ~207 of page 9 would be helpful.

We believe the reviewer is referring to text relating to CNA differences being more attributable to CN losses in MEL/NPLAS samples than to CN gains in MES samples. All MEL/MES genes are depicted graphically in **Supplementary Data 1-4** which include per-sample CNA. At a group-wise summary level, CNA (gains or losses) are indicated in the multi-modal plots shown in **Figure 6a and b** which provide the critical high-level view that best supports our conclusion.

4. What are the biological implications for limited positive correlation between epigenetic age and chronological age seen in INT? Statistical p value needs to be presented with slope or rho value.

As described in the text, DNAmAge was reported by its original creators to be unreliable in cancer tissues, consistent with an accelerated and partly stochastic trajectory of genomic damage being

integral to the process of carcinogenesis. Nonetheless, the apparently distinct DNAmAge-chronological age association for the NPLAS subgroup suggests that a different underlying process of methylation alteration is at play in the NPLAS phenotype. Whether this has specific inherent biological implications is not yet known, but we may speculate that it could have implications for methylation/demethylation-based therapeutics. However, we feel this is too highly speculative to raise further in this manuscript. Statistics have been included as requested.

Reviewer #3 (Remarks to the Author):

Andrews et al. generated melanoma cell lines from patient-derived metastatic melanoma tumors and these cells were used for their multi-modal analyses with the combination with publicly available dataset. Firstly, authors classified melanoma cells into $K = 3$ based on the results of initial analyses. The melanoma cell transcriptomic states (MCS) are categorized as MEL (melanocytic), INT (intermediate) and those MCS showed specific gene expressions such as MITF, SOX10, and EDNRB for MEL, ZEB1, AXL, and EGFR for MES. Additionally, MCS are further investigated by using multiple omics data such as DNA methylation, CNA, miRNA, and lncRNA.

Here are my comments.

Major comments

1. RNA expression levels were normalized by RPKM and which is known to be inconsistent among samples (Bioinformatics. 2010, 26(4):493-500, Theory Biosci. 2012, 131(4):281-285 etc). It is highly suggested to use TPM for expression analysis. Actually, lncRNA was normalized to have TPM values in your study.

We very much appreciate the reviewer's suggestion to use TPM for expression analysis and have re-run all expression analyses using TPM summarized data.

2. In Figure 2b and c, authors conducted t-test after ANOVA, which is incorrect. Please do appropriate post-hoc test whether the difference is statistically significant.

Thank you for identifying this error. We have applied Kruskal-Wallis tests across clusters together with Benjamini-Hochberg correction for multiple comparisons (due to multiple deconvoluted cell types).

3. What is the precise definition of early-passage melanoma cell lines? Please describe more detailed in method section.

We take our guidance on low or early-passage from the American Type Culture Collection (ATCC)¹ and the European Collection of Authenticated Cell Cultures (ECACC)² which through their published experimental results and general guidelines consider ten or fewer passages to be "low passage." For example, the ECACC guidelines state, "... only perform a relatively low number of passages, e.g. 10 – 20, before returning to another ampoule of the same stock." All cell lines used in this study were 10 or fewer passages, and wording in the Methods has been clarified in relation to this.

¹ <https://www.atcc.org/resources/technical-documents/passage-number-effects-in-cell-lines>

² <https://www.phe-culturecollections.org.uk/technical/cell-line-passage-numbers-explained.aspx>

4. Some data (both main figures and supplementary figures) such as mutational signature analysis shown in Figure 3d bottom for example are not explained in the manuscript. Please explain or omit if unnecessary.

We have ensured that all data presented graphically is described in legends and/or main text.

5. P14, line 322: The authors stated, "However, yet further layers of functional control are indicated, as evidenced by low correlation between EGFR transcript levels and EGFR phosphorylation at functionally critical tyrosine residues determined by RPPA (pY1068 Pearson rho 0.44, $p < 0.001$; pY1173 Pearson rho -0.019, $p = 0.55$; Fig. S8e-f)". It will be difficult to draw a conclusion based on the data in Fig. S8e-f because many studies demonstrated that there is an expression gap between RNA expression and protein expression. Therefore, correlation analyses between RNA expression and phospho-proteins are misleading.

The reviewer's point is well taken. We did not intend to suggest a generalizable correlation between transcript and phosphoprotein expression but appreciate the potential for the reader to assume so, which would indeed be misleading, thus we have removed Fig. S8e-f to avoid confusion.

6. P9, line 194, Author stated, "a higher frequency of BRAF hotspot events in the MEL samples ($p = 0.016$, Fisher's exact test) and of NRAS hotspot events in the MES samples ($p = 0.03$, Fisher's exact test), consistent with the typically mutually exclusive alteration of these two genes." I am curious to know the expression levels of those two genes. In other words, genes shown in Fig 3a-c are not frequently mutated judging from Fig 3d. How would you interpret these gaps between expression levels and gene mutations? One explanation could be as author stated in line 204, "CNA differences were observed between MCS, e.g. gain of chromosomes 1p, 7q, 15 (p+q), and losses of chromosomes 9p, 11 (p+q), 18p, 19q (Fig 3e)." However, MITF locates on Chr3, ZEB1 on Chr10, AXL on 19, SOX10 on Chr22, EDNRB on Chr13, and EGFR on Chr7p. None of the genes of interest mapped to the indicated gene locus, indicating that those gene expression levels might be regulated by other mechanisms, not

CNA. This suggests that it should be indirect regulation, as TRPM1 and miR-211 locate on Chr15. In addition, authors stated in line 197, "In high-purity TCGA melanoma samples, BRAF mutations were not associated with MCS, potentially reflecting an interaction between BRAF status and MCS regarding adaptability to grow in vitro, even at low passages (Fig. S4)." If you conclude abovementioned, what is the advantage of using TCGA data in your analyses?

We agree with the reviewer that the MCS gene expression profiles do not align with recurrently mutated genes or mutations in the MCS genes themselves. This is consistent with prior data demonstrating that the small number of commonly recurrent somatic mutations (principally *BRAF* and *NRAS*) in melanoma are early tumorigenic events in the clonal lineage of tumors or even in pre-malignant naevi (e.g. Shain *et al.*, NEJM 2015, doi: 10.1056/NEJMoa1502583), upon which different gene expression profiles and cellular functions occur (e.g. Widmer *et al.*, Pigment Cell Melanoma Res. 2012, doi: 10.1111/j.1755-148X.2012.00986.x).

The reviewer rightly points out that many of the genes that have been identified in the field to be of functional consequence do not align with the arm-length SCNAs we observe to be significantly different between MEL and MES cells, indicating that SCNA alone is likely not sufficient to explain low or high transcript expression levels of MCS genes. A key point of the paper is that our analysis reveals how multiple molecular features (SCN gains/losses at the allelic level, miRNA transcript targeting, DNA gene methylation) tend to align in a manner consistent with the expression status of the MEL or MES genes, revealing the complex coordination of defined molecular features that underlie the MEL versus MES gene expression patterns. Certainly, we observe certain known direct gene-level amplification/gene expression events (e.g. *MITF* is known to be amplified in a small subset of melanomas, and we observe that our MEL cell lines S2400, S2279 and others are clearly *MITF* amplified), but these events only account for a small number of cases and for some genes.

Supplementary Data 1 – MITF excerpt:

We thus also entirely agree with the reviewer that indirect regulation contributes to the observed transcriptomic features. The reviewer points out that the *TRPM1* and *miR-211* genes, which display differentially heightened gene expression in MEL cells localize to chromosome 15q, which is the most significant arm-level copy number gain in MEL cells. Given our observations that 1) *miR-211* is

upregulated and inversely correlated with *ZEB1* in MEL cells and 2) that miR-211 is known to target *ZEB1* mRNA, we tested whether miR-211 modulated *ZEB1* in melanoma cells. Exogenous expression of miR-211 in cells with elevated *ZEB1* resulted in a marked reduction of *ZEB1* at the transcript and protein levels. These results are presented and discussed in **response to Reviewer 1 Question 3** above.

7. Does the MEL-MES state reflect the transition from MEL to MES, from MES to MEL, or truly mixture of both? P6, line 125, authors stated, "Cluster 2 was described as intermediate (INT), consistent with a transitional state in the phenotype-switching model of melanoma previously by others." If this is the case, does the MEL-MES state reflect the transition from MEL to MES, from MES to MEL, or truly mixture of both? However, author also stated in P10, line 216, "As in our own line, CNA profiles indicated the existence of a distinct INT state rather than these samples representing a transitional or mixed state." On top of that, P11, line 244, "Despite high data dispersion, linear regression revealed highly similar trends for both MEL and MES groups for which DNAmAge was nearly constant regardless of chronological age (slope = -0.0023 MEL, -0.023 MES). By contrast, epigenetic age of INT samples showed positive correlation with chronological age (slope = 0.50). Together with distinct CNA profiles, these findings add further support to the existence of fundamental molecular differences between the INT and either MEL or MES states." Initially author described the INT state as "transitional state". Later, described as it is not. Please explain how you would interpret logically.

We apologize for the confusion regarding the nature of the non-MEL/non-MES state which is likely in large part due to a lack of characterization of this state in our manuscript and a poor choice of terminology referring to it as the "intermediate" state. We have now performed additional analyses and comparisons with other published signatures as suggested by Reviewer 1 in order to characterize this state further on its own merits, and accordingly have re-named it a "neural-plastic" (NPLAS) state.

Regarding its interpretation, we present several lines of evidence suggesting that it is a more independent state than suggested by the relatively linear interpretation of the "phenotype switching" model which places melanoma cells along a spectrum between proliferative and invasive extremes. Several studies, including those analyzed in response to Reviewer 1, posit multiple states which also appear to diverge from this linear trajectory. In so far as each state may represent a molecular "attractor" state displaying a degree of phenotypic stability, they could have implications for therapeutic sensitivity although whether any given state is sensitive or resistant to therapy would be expected to vary depending on the mechanism of action of the therapy being considered.

We have adjusted wording and terminology throughout the text in order to clarify our interpretation of the MCS.

8. Supplementary Figure S9e-f, which groups showed a statistically significant? How did you analyze to obtain p-values?

We have generally applied Kruskal-Wallis test for significance across comparisons involving all three clusters, as indicated in the figure legends as few situations warrant post-hoc pairwise comparisons upon inspection of the spread of data. The data relating to macrophage and cytolytic signatures in the Riaz cohort (now **Supplementary Figure 9d-e**) now also indicate post-hoc Dunn test results.

Minor comments

1. There is no color represented in Supplementary Figure S3A, color key box. Please fix it.

This figure has been updated.

2. Format; please carefully review to amend the manuscript, e.g. *p* has to be *p* (italic format) and it should be shown with a following blank like $p = 0.01$ in Nature communications. There are two descriptions, one is 13 and the other is thirteen, for another example. Please amend all of them.

We have unified formatting as much as possible but will certainly address any journal-specific requirements as appropriate during a future proofing process.

3. Why do the authors proceed with low RIN samples (2350, 2417, 2734) for miRNA sequencing? Is there any specific reason why authors did not redo extracting RNAs?

In the effort to reduce potential technical artifacts in the analysis, we performed sequencing of the samples in large batches. Given that miRNAs are extremely stable against nucleases, RIN is not generally considered a good indicator of miRNA degradation (Jung *et al.*, Clinical Chemistry 2010, doi: 10.1373/clinchem.2009.141580), and we proceeded with such samples for miRNA analysis. Unsupervised analysis of these samples did not indicate any outlier effects and no sequencing read drop off was observed for miRNAs in these samples.

4. Molecular insight to the metastatic melanoma is weak. Authors performed RNA-expression analysis, DNA methylation analysis, GO analysis and so on but ended up in a possibility. Since authors generated cell lines, all or at least some of the important findings described in the manuscript should be confirmed by experimentally. For instance, P12, line 23, author stated, "While high transcript levels of ZEB1 in MES cells were associated with DNA methylation at gene body site, comparable of methylation in INT samples may be moderated by lower levels of LINC00702 in these cells, leading to increased availability of the ZEB1-targeting miRNA-4652-3p, thus reducing ZEB1 expression 56 (Fig. S7d, Fig. 4c)." It is good to validate the miRNA expression levels by northern blotting and protein expression levels by western blotting, or other reasonable methodologies.

We appreciate the Reviewer's emphasis on validating some of the important observations made in the paper. The correlation of particular lncRNAs with either MEL or MES gene expression profiles was intended to be a minor aspect of the current manuscript which we simply wanted to include as part of a comprehensive data resource at this time. We now see that inclusion of results from reference 56 which showed LINC00702 to act as a ceRNA sponge of miR-4652-3p to release the expression of ZEB1 in a non-melanoma tumor type, went well beyond our intent. We have removed reference 56 and have emphasized the point that inclusion of the lncRNA correlation in the supplemental data is meant as a resource for future focused experimental analysis.

An observation we believe to be very relevant to MEL versus MES gene expression and phenotype differences is the strong anti-correlation between miR-211 and ZEB1. Both miR-211 and ZEB1 are well-described melanoma-associated genes. miR-211 has been demonstrated to be a target of MITF (melanocyte inducing transcription factor), a master regulator of melanocyte differentiation, and is elevated in MEL-like melanomas, which express the pigmentation apparatus associated with melanocyte differentiation and melanoma initiation. Conversely, there is low or no expression of miR-211 in MES-like melanoma cells and tumors. Multiple prior experiments have shown that exogenous expression of miR-211 in non-pigmented, highly invasive, melanoma cells blunted their invasive phenotype, whilst targeted reduction of miR-211 in pigmented, highly reproducing, melanoma cells reduced their growth rate (e.g. Yu and Yang, Biochem Biophys Res Commun 2016;476(4):400-405; Mazar *et al.*, PLoS One 2010 5(11):e13779; Levy *et al.*, Mol Cell 2010;40(5):841-9). The inverse associations between elevated ZEB1 and invasion and low ZEB1 and cell growth are also well-documented (e.g. Caramel *et al.*, Cancer Cell 2013;24(4):466-80; Denecker *et al.*, Cell Death Differ 2014;21(8):1250-61). Given this background information, the strong degree of anti-correlation between miR-211 and ZEB1 observed in the cell line set and TCGA tissue cohorts, and the predictive miR-targeting analysis performed, we focused our validation on these two features. We performed PCR-based analysis of cell lines with low versus high levels of miR-211 and observed inverted levels of ZEB1 by this method similar to the RNA-Seq analysis. As requested by the

reviewer, we performed a correlation with both ZEB1 transcript and protein levels, which showed a strong association between transcription and translation products. Exogenous expression of miR-211 in melanoma cell lines with high ZEB1 markedly reduced ZEB1 transcript and corresponding protein levels compared to control. These results have also been shown and discussed in the **response to Reviewer 1 Question 3** above.

5. As this paper mentioned (BMC cancer, 2015; 15: 242), there are no standardized log₂ ratio cut-offs to define low-amplitude copy number gain/loss and high amplitude amplification/deletion. Based on the available published literature, author can consider to use a log₂ ratio cut-off of +/- 0.25 to define copy number gain/loss and a cut-off of +/- 0.8 to define amplification and deletion. Please double-check the term; loss or deletion, gain or amplification. Related that, please amend the methods section "Somatic copy number alteration and purity analysis"

The reviewer is right to point out that there is no standard log₂ ratio cut-offs for low- vs. high-amplitude gain/loss or deletion/amplification. A log₂ ratio of +/- 0.5 was used throughout the study as a cut-off for copy number altered assuming a diploid genome as normal. We have now made this explicit in the methods section and have standardized the language throughout the text.

6. Please state in the manuscript when publicly available dataset was downloaded and used for analyses, e.g. TCGA-SKCM and others as author did state in CCLE dataset (accessed October 18, 2020).

For TCGA SCM data we used the Broad Firehose 2016 version (stable legacy release).

For the Riaz dataset we obtained raw count data from the github repository at https://github.com/riazn/bms038_analysis/tree/master/ (accessed 10/17/20).

These details have been added to the methods.

7. Please give more details of both random forest and support vector machine's hyper parameter settings as well as obtained results. Also, please describe the result of SVM performance since none of the information is available in the main text but stated in the methods section.

Due to the poor performance of SVM in preliminary testing this was neither carried forward nor repeated/reattempted with the new TPM-based dataset thus we have removed the mention of SVM from methodology.

With the re-processing of data starting with TPM input and reduced number of published immune checkpoint inhibitor datasets utilized during revision of the manuscript, we have also revised the strategy for MEL/MES classifier random forest model development, focusing on a model trained on our cell line data, given that this is most concordant with the principle of addressing melanoma cell-intrinsic phenotypes. Due to the modest sample size, the model was trained on all MDACC cell lines rather than a split training and testing subset.

We have updated the relevant methods section to include details of the methodology used, which includes pre-processing to remove highly correlated genes, down-sampling, and selection of the lowest number of predictors providing accuracy within an acceptable degree of variation from the maximum using the 'tolerance' method.

Reviewers' Comments:

Reviewer #1:

Remarks to the Author:

Andrews et al. have fully addressed all my points as well as the comments from the other reviewers. The revised version of the manuscript is substantially improved and, in my opinion, reaches the relevance and scientific standards for publication in Nature Communications. I'd suggest the following edits:

1.- Different names are used for the MCS gene sets (e.g "Andrews et al." in Fig S1d, "MDACC" in Fig S1g or "MDACC MEL/MES" in Fig S2a. Try to use consistent nomenclature throughout the manuscript and clarify figure legends. Specially, when gene sets are referred as "MEL/MES" it is not clear if only clusters 1 and 3 or all 1-3 are included.

2.- I appreciate the large amount data presented in the manuscript. However, given the pleiotropic and context-dependent functions of miRNAs, the experimental validation of Zeb1 regulation by miR-211 should be mentioned and if possible, Figure R1Q3 included in the manuscript. Direct regulation of Zeb1 expression by miR-211 was found in cervical cancer but not in melanoma, where regulation of other genes involved in invasion was demonstrated (Ray et al., 2021, doi:10.3389/fonc.2020.628367). Therefore, these results strongly support the -omics associations and interpretation by the authors.

3.- I think it'd make more sense to describe the sensitivity of the cell lines to BRAF inhibitor (now Fig S1f) in the context of the therapeutic implications of the MCS (i.e., at the end of section "Clinical outcomes for melanoma sample donors are influenced by MCS"). That'd support the trends observed from the clinical data.

Reviewer #2:

Remarks to the Author:

Authors provide a comprehensive response to reviewer comments, albeit, I find them limited. Despite the modifications made in the course of revising this manuscript, data in support of authors finding is lacking, and as such, this manuscript is rather speculative.

Reviewer #3:

Remarks to the Author:

Basically, I feel that the author has effectively addressed the criticisms from me that I pointed out in my first review. I would recommend publication.

REVIEWERS' COMMENTS

Reviewer #1 (Remarks to the Author):

Andrews et al. have fully addressed all my points as well as the comments from the other reviewers. The revised version of the manuscript is substantially improved and, in my opinion, reaches the relevance and scientific standards for publication in Nature Communications. I'd suggest the following edits:

1.- Different names are used for the MCS gene sets (e.g "Andrews et al." in Fig S1d, "MDACC" in Fig S1g or "MDACC MEL/MES" in Fig S2a. Try to use consistent nomenclature throughout the manuscript and clarify figure legends. Specially, when gene sets are referred as "MEL/MES" it is not clear if only clusters 1 and 3 or all 1-3 are included.

We thank reviewer 1 for their supportive comments. As requested, we have unified and/or clarified nomenclature throughout our manuscript. Importantly, when referring to the cell line cohort, we have used "MDACC cell lines", when referring to the gene sets derived from differential gene expression analyses of our cell line cohort we have used "MEL" to indicate genes enriched in the MEL samples (cluster 1), "MES" to indicate genes enriched in the MES samples (cluster 3), and avoided the use of "MEL/MES" in favor of "MEL or MES" or "MEL and MES" to be more explicit when referring to both gene sets. We have indicated when the use of MEL, MES or NPLAS refers to genes, a cellular state, or samples classified within that cellular state, as appropriate throughout the text.

2.- I appreciate the large amount data presented in the manuscript. However, given the pleiotropic and context-dependent functions of miRNAs, the experimental validation of Zeb1 regulation by miR-211 should be mentioned and if possible, Figure R1Q3 included in the manuscript. Direct regulation of Zeb1 expression by miR-211 was found in cervical cancer but not in melanoma, where regulation of other genes involved in invasion was demonstrated (Ray et al., 2021, doi:10.3389/fonc.2020.628367). Therefore, these results strongly support the -omics associations and interpretation by the authors.

We appreciate the suggestion and have added both text and figures to the main manuscript. miR-211/ZEB1 related figures now appear in main Figure 5 panels d and e.

3.- I think it'd make more sense to describe the sensitivity of the cell lines to BRAF inhibitor (now Fig S1f) in the context of the therapeutic implications of the MCS (i.e., at the end of section "Clinical outcomes for melanoma sample donors are influenced by MCS"). That'd support the trends observed from the clinical data.

These results have been adjusted as suggested. The cell line BRAF inhibitor sensitivity data now appears as Supplementary Figure 9b, and remaining panels of this figure have been moved to a new Supplementary Figure 10a-d.

Reviewer #2 (Remarks to the Author):

Authors provide a comprehensive response to reviewer comments, albeit, I find them limited. Despite the modifications made in the course of revising this manuscript, data in support of authors finding is lacking, and as such, this manuscript is rather speculative.

We thank reviewer 2 for taking the time to examine our comprehensive responses and revisions, which we feel do strengthen our manuscript. As a primarily integrative analysis of data generated from multiple platforms, we agree that further work is required to functionally validate and extend our findings towards clinical application, but nonetheless present data that are of clear interest to the readership and will stimulate robust translational development.

Reviewer #3 (Remarks to the Author):

Basically, I feel that the author has effectively addressed the criticisms from me that I pointed out in my first review. I would recommend publication.

We thank reviewer 3 for their excellent prior recommended edits which have contributed to a much-improved overall manuscript.